# PERFORMANCE BOUNDS FOR MODEL AND POLICY TRANSFER IN HIDDEN-PARAMETER MDPS

**Haotian Fu[1], Jiayu Yao[2], Omer Gottesman[1], Finale Doshi-Velez[2] & George Konidaris[1]**
[1]Brown University, [2]Harvard University

## ABSTRACT

In the Hidden-Parameter MDP (HiP-MDP) framework, a family of reinforcement learning tasks is generated by varying hidden parameters specifying the dynamics and reward function for each individual task. The HiP-MDP is a natural model for families of tasks in which meta- and lifelong-reinforcement learning approaches can succeed. Given a learned context encoder that infers the hidden parameters from previous experience, most existing algorithms fall into two categories: *model transfer* and *policy transfer*, depending on which function the hidden parameters are used to parameterize. We characterize the robustness of model and policy transfer algorithms with respect to hidden parameter estimation error. We first show that the value function of HiP-MDPs is Lipschitz continuous under certain conditions. We then derive regret bounds for both settings through the lens of Lipschitz continuity. Finally, we empirically corroborate our theoretical analysis by varying the hyper-parameters governing the Lipschitz constants of two continuous control problems; the resulting performance is consistent with our theoretical results.

## 1 INTRODUCTION

Hidden-parameter Markov Decision Processes (HiP-MDPs) (Doshi-Velez & Konidaris, 2016) describe a family of related tasks by modeling task variations with a set of low-dimensional hidden parameters. HiP-MDPs are widely used in recent meta-Reinforcement Learning (RL) and lifelong RL works, in which an agent needs to quickly adapt to new tasks by transferring the knowledge from previous tasks. To solve HiP-MDPs, most existing algorithms first infer the hidden parameters from previous experience (e.g. by learning a context encoder that maps trajectories to a hidden parameter estimate $\theta$), then use the estimated hidden parameters to solve new tasks with either *model* (Killian et al., 2017; Lee et al., 2020; Fu et al., 2022) or *policy* transfer (Yao et al., 2018; Rakelly et al., 2019) algorithms. As shown in Figure 1, in model transfer, the inferred hidden parameters are used to build a simulator of the environment, which can then be used for planning. In policy transfer, the inferred hidden parameters are used to parameterize the policy of the new task directly. Previous works have observed mixed empirical evidence for when each approach performs better (Yao et al., 2018; Lee et al., 2020; Fu et al., 2022). In this work, we take a step in the theoretical direction by studying the regret bounds of model and policy transfer algorithms, respectively, which helps characterize the robustness of these algorithms.

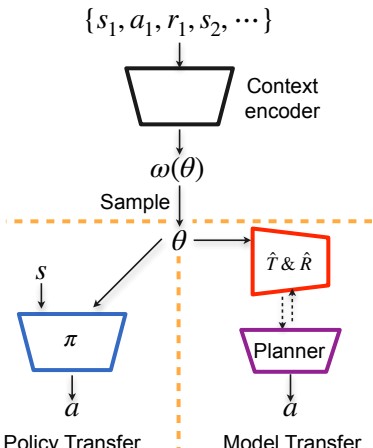

Figure 1: Difference between model and policy transfer in HiP-MDPs.

Two main factors affect the performance of HiP-MDP algorithms: (1) the estimation accuracy of the hidden parameter used as an input to either the learned model or policy, and (2) the quality of the learned model or policy (i.e., given an accurately estimated hidden parameter, whether the agent can reach the optimal performance). In practice, the HiP-MDP tasks considered in most recent meta-RL papers (e.g., changing the physical properties of MuJoCo-simulated robots) are relatively easy to

solve when the hidden parameters are given. We therefore assume the learned model and policy are optimal and theoretically analyze how hidden parameter estimation error affects the final performance of model and policy transfer algorithms.

Our contributions are as follows. We first derive the conditions under which the value function of HiP-MDPs is Lipschitz continuous. We then derive upper bounds for regret in model and policy transfer algorithms, respectively, when the estimation error of the hidden parameters is bounded. We also give an upper bound for multi-step prediction error in model transfer. We further prove that the derived bounds are tight when the transition dynamics, reward function and policy are linear and deterministic. As far as we are aware, these are the first theoretical results about policy and model transfer performance bounds in HiP-MDPs. Given the same hidden parameter estimation error bound, our results characterize when model or policy transfer can be more robust than the other: a slower increase in the Lipschitz constant with respect to the hidden parameter implies more robustness.

In addition to theoretical analysis, we empirically study the performance of model and policy transfer algorithms in two continuous control domains. For each domain, we control the Lipschitz constants of the HiP-MDPs by altering the hyper-parameters of the environments. The results are consistent with our theoretical understanding of how the regrets of model and policy transfer algorithms scale with respect to the estimation error of the hidden parameters: a slower increase in Lipschitz constant with respect to the hidden parameter implies a smaller performance decay.

## 2   BACKGROUND AND RELATED WORK

HiP-MDPs model the variations in the transition dynamics and reward functions by assigning each task a hidden parameter $\theta$, drawn from the distribution $P_\Omega$. The agent neither observes $\theta$ nor has access to the the distribution $P_\Omega$ that generates the task family. For a given task, parameterized by $\theta \in \Theta$, the stochastic dynamics are given by $T(s'|s, a; \theta)$ and the deterministic reward function by $R(s, a; \theta)$. We consider continuous state and action space in this work ($s \in S, a \in A$). Upon encountering a new task, the agent estimates the new dynamics, $\hat{T}$, and the new reward function, $\hat{R}$, by inferring a distribution $\hat{\omega}(\theta)$ over the hidden parameter. If our estimation of $\theta$ is accurate, then $\hat{\omega}(\theta)$ should be close to the true distribution $\omega(\theta)$, i.e. peaking at the true $\theta$ for this specific task. We assume the learned dependency of $\hat{T}$ and $\hat{R}$ on $\theta$ is accurate. Thus, if the estimation of $\theta$ is also accurate, the agent can solve the task completely.

The HiP-MDP is an important setting widely used in recent meta/lifelong RL papers (see Appendix G). A number of approaches have been proposed to infer the hidden parameter $\theta$ of HiP-MDPs in the latent representation space, such as using Bayesian models to leverage prior knowledge (Killian et al., 2017; Yao et al., 2018; Fu et al., 2022), or training a context encoder that maps trajectories to the latent parameter (Rakelly et al., 2019; Zintgraf et al., 2020; Lee et al., 2020).

Given the inferred hidden parameters, various RL algorithms have been used to adapt to the new tasks. The downstream RL algorithm given the inference of the hidden parameter includes training an off-policy actor-critic that takes in the estimated parameter as an additional input (Rakelly et al., 2019; Fakoor et al., 2020; Fu et al., 2021), or leveraging recurrent neural networks (Zintgraf et al., 2020; Duan et al., 2016), or planning based on the learned transition model that also takes in the inferred parameter (Lee et al., 2020; Mendonca et al., 2020). Yet, in practice, there is no clear winner among these algorithms, leaving the question that what factors affect the performance of different algorithms, and how. Nair & Doshi-Velez study a similar problem but in a different setting where the hidden parameters are given (contextual MDP) and derive sample complexity bounds for model-based learning. In this work, we quantify how errors and uncertainty in estimating $\theta$ affect the performance of model and policy transfer methods through the lens of Lipschitz continuity.

Given a distance metric on the space $M$, $d_M$, Lipschitz continuity quantifies the smoothness of a function as follows.

**Definition 2.1.** A function $f \colon M_1 \mapsto M_2$ is uniformly Lipschitz continuous if

$$K_f^{M_1, M_2} := \sup_{x_1, x_2} \frac{d_{M_2}(f(x_1), f(x_2))}{d_{M_1}(x_1, x_2)}. \tag{1}$$

We use $K_f^{M_1}$ in some places for simplicity of notation. Throughout the work, we use the Euclidean distance for $d$ in the case of point vectors and the Wasserstein distance in the case of probability distributions. The main goal of this work can now be stated as deriving the relation between $d(\omega(\theta), \hat{\omega}(\theta)) \leq \epsilon$, and the performance of model transfer and policy transfer methods.

The Wasserstein Metric can be linked to Lipschitz continuity using duality (Villani, 2008):

$$W(\omega_1, \omega_2) = \sup_{f:K_f^{M_1,M_2} \leq 1} \int (f(x)\omega_1(x) - f(x)\omega_2(x))dx. \tag{2}$$

Given a distribution $\omega(\theta)$ over the hidden parameters, we define the generalized transition, reward and value functions in HiP-MDPs as:

$$T_g(s'|s, a, \omega) = \int_\theta T(s'|s, a; \theta)\omega(\theta)d\theta, \quad R_g(s, a, \omega) = \int_\theta R(s, a; \theta)\omega(\theta)d\theta,$$

$$V_g(s, \omega) = \int_\theta V(s, \theta)\omega(\theta)d\theta, \quad Q_g(s, a, \omega) = \int_\theta Q(s, a, \theta)\omega(\theta)d\theta.$$

These generalised functions are an expectation over the uncertainty in the hidden parameter. All the $Q$ and $V$ functions we use below refer to the generalized function $Q_g$ and $V_g$.

We are now equipped to formally state all the smoothness assumptions we make about the HiP-MDPs:

**Definition 2.2.** A HiP-MDP is $(K_R^S, K_T^S, K_R^A, K_T^A, K_R^\Omega, K_T^\Omega)$-Lipschitz if, for all $\omega \in \Omega$, and s $\in S$ and a $\in A$:

$$|R_g(s_1, a, \omega)) - R_g(s_2, a, \omega))| \leq K_R^S d_S(s_1, s_2), W(T_g(\cdot|s_1, a, \omega)), T_g(\cdot|s_2, a, \omega))) \leq K_T^S d_S(s_1, s_2)$$

$$|R_g(s, a_1, \omega)) - R_g(s, a_2, \omega))| \leq K_R^A d_A(a_1, a_2), W(T_g(\cdot|s, a_1, \omega)), T_g(\cdot|s, a_2, \omega))) \leq K_T^A d_A(a_1, a_2)$$

$$|R_g(s, a, \omega_1) - R_g(s, a, \omega_2)| \leq K_R^\Omega d_\Omega(\omega_1, \omega_2), W(T_g(\cdot|s, a, \omega_1), T_g(\cdot|s, a, \omega_2)) \leq K_T^\Omega d_\Omega(\omega_1, \omega_2)$$

To simplify notation, we will write that a HiP-MDP is Lipschitz continuous when all the constants can be implied from the context.

In the following theoretical analysis, we assume the HiP-MDPs are Lipschitz. Intuitively, this implies that when the hidden parameters of two MDPs are close, their transition dynamics and reward functions are similar, and for a given task, transitions dynamics and rewards do not change much given small changes in the states and actions. Similar assumptions have commonly been made in previous works (Hinderer, 2005; Asadi et al., 2018; Gottesman et al., 2021; Gelada et al., 2019; Ren et al., 2019; Luo et al., 2019). We extend the smoothness assumption to the hidden parameters for a better understanding about the role of hidden parameter estimation error in the final performance.

In addition to the smoothness assumptions about the HiP-MDPs, we make the following assumptions about the smoothness of policies.

**Definition 2.3.** A policy $\pi(a|s, \omega(\theta))$ is $(K_\pi^S, K_\pi^\Omega)$-Lipschitz continuous if:

$$K_\pi^S := \sup_\omega \sup_{s_1, s_2} \frac{W(\pi(\cdot|s_1, \omega(\theta)), \pi(\cdot|s_2, \omega(\theta)))}{d_S(s_1, s_2)},$$

$$K_\pi^\Omega := \sup_s \sup_{\omega_1, \omega_2} \frac{W(\pi(\cdot|s, \omega_1(\theta)), \pi(\cdot|s, \omega_2(\theta)))}{W(\omega_1(\theta), \omega_2(\theta))}$$

are finite.

We evaluate policies by calculating their expected discounted reward given a distribution of initial states, $\mu(s)$:

$$J_\pi^\mu := \int_s V_g^\pi(s, \omega(\theta))\mu(s)ds. \tag{3}$$

Lipschitz continuity in MDPs has been studied by many previous works (Hinderer, 2005; Rachelson & Lagoudakis, 2010; Tang et al., 2020; Gottesman et al., 2021). Pirotta et al. (2015) propose to use Lipschitz continuity property of MDPs to speed up policy gradient algorithms. Asadi et al. (2018) show how the magnitude of the Lipschitz constant of the transition dynamics affects the performance of model-based RL algorithm's performance is influenced by the magnitude of the Lipschitz constant of the model. Gelada et al. (2019) leverages Lipschitz continuity assumptions in regular MDPs to learn state abstractions. However, none of them conduct theoretical analysis on HiP-MDPs.

# 3    LIPSCHITZ VALUE FUNCTION OF HIP-MDPS

Our first main result is showing under which conditions the value functions are Lipschitz continuous for HiP-MDPs. We show this through a combination of Theorem 3.1 and Theorem 4.3 & 5.2 in the following sections. Theorem 3.1 mainly studies the Lipschitz continuity with respect to $S$ and $A$ while the hidden parameter is fixed. This can be seen as a generalization of theorems derived in prior papers (Rachelson & Lagoudakis, 2010; Pirotta et al., 2015; Asadi et al., 2018; Gelada et al., 2019) that study Lipschitz value function for a regular MDP.

**Theorem 3.1.** *For a $(K_R^S, K_T^S, K_R^A, K_T^A, K_R^\Omega, K_T^\Omega)$-Lipschitz HiP-MDP with a $(K_\pi^S, K_\pi^\Omega)$-Lipschitz policy $\pi$, if $\gamma(K_T^S + K_\pi^S K_T^A) < 1$,[1] the value function is Lipschitz continuous with respect to $S$ and $A$ with constants bounded by:*

$$K_{Q^\pi}^A := \sup_\omega \sup_s \sup_{a_1, a_2} \frac{|Q(s, a_1, \omega) - Q(s, a_2, \omega)|}{d_A(a_1, a_2)} \leq \frac{K_R^A - \gamma(K_R^A K_T^S - K_R^S K_T^A)}{1 - \gamma(K_T^S + K_\pi^S K_T^A)}$$

$$K_{Q^\pi}^S := \sup_\omega \sup_a \sup_{s_1, s_2} \frac{|Q(s_1, a, \omega) - Q(s_2, a, \omega)|}{d_S(s_1, s_2)} \leq \frac{K_R^S - \gamma K_\pi^S (K_R^S K_T^A - K_R^A K_T^S)}{1 - \gamma(K_T^S + K_\pi^S K_T^A)}$$

$$K_{V^\pi}^S := \sup_\omega \sup_a \sup_{s_1, s_2} \frac{|V(s_1, \omega) - V(s_2, \omega)|}{d_S(s_1, s_2)} \leq K_{Q^\pi}^S + K_{Q^\pi}^A K_\pi^S.$$

*Proof.* (sketch) We derive the value difference bound by decomposing it into the combination of the Q-value difference with respect to $S$ & $A$ using our previous definitions and the Bellman update rule (9). Then we get the bounds for $K_{Q^\pi}^A$ and $K_{Q^\pi}^S$ respectively by applying the dual form of the Wasserstein Metric (Equation 2) and with the fixed-point iteration. See appendix E for all proofs.    □

The above theorem shows the conditions under which the value functions of HiP-MDPs are Lipschitz continuous with respect to states and actions. We can get the following corollary when the policy is optimal:

**Corollary 3.2.** *If the policy $\pi$ is optimal and $\gamma K_T^S < 1$, then:*

$$K_{Q^*}^S \leq \frac{K_R^S}{1 - \gamma K_T^S}, \quad K_{Q^*}^A \leq \frac{K_R^A + \gamma K_R^S K_T^A - \gamma K_T^S K_R^A}{1 - \gamma K_T^S}.$$

Note that in contrast to $K_{Q^*}^S$, $K_{Q^*}^A$ is affected by the smoothness of the transition dynamics and reward functions with respect to action space ($K_T^A$ and $K_R^A$).

Theorem 3.1 and Corollary 3.2 also directly apply to a $(K_R^S, K_T^S, K_R^A, K_T^A)$-Lipschitz **regular** MDP with a $K_\pi^S$-Lipschitz policy, as the Lipschitz continuity assumptions with respect to hidden parameters are not used in both the final results and proofs. Specifically, the upper bound for $K_{Q^*}^S$ in Corollary 3.2 recovers the results in Asadi et al. (2018); Gelada et al. (2019). If we assume the Lipschitz constants for state and action are the same, we get Corollary E.1, which recovers the results in Rachelson & Lagoudakis (2010); Pirotta et al. (2015). For **regular** MDPs, our results are more general compared to prior work as (1) we do not assume that the Lipschitz constants for state and action are the same because that is usually not the case in practice; (2) our theory applies to all Lipschitz policies rather than only the optimal policy.

One of our major goals in this work is to derive the conditions for Lipschitz value functions of HiP-MDPs. Theorem 3.1 shows that the value functions in HiP-MDPs are Lipschitz continuous with respect to $S$ and $A$ under certain conditions. To complete the theorem for Lipschitz value function in HiP-MDPs, we must also explore the conditions under which the value function is Lipschitz continuous with respect to the remaining input of Q-function in HiP-MDPs, $\omega$. As we shown in the next two sections, exploring this actually results in different bounds for policy transfer and model transfer. By leveraging the results of Lipschitz value functions, we then find how the estimation error $d(\omega(\theta), \hat{\omega}(\theta)) \leq \epsilon$ of the hidden parameter at test time will affect the performance of model and policy transfer algorithms respectively. We quantify the performance gap by deriving the upper bound of the regret as a function of the error in estimating the hidden parameter. Note that throughout the following analysis, we abstract out the discussion of how $\hat{\omega}(\theta)$ is estimated, and assume it is given with an estimation error bounded by $\epsilon$.

---

[1]We make this assumption in many of our following theorems. See appendix F for explanations.

## 4 PERFORMANCE BOUNDS OF MODEL TRANSFER IN HIP-MDPS

We now focus on the regret of model transfer in HiP-MDPs induced by the hidden parameter estimation error, $d(\omega(\theta), \hat{\omega}(\theta)) \leq \epsilon$. We assume that the agent has learnt an accurate transition function $\hat{T}_g(s'|s, a, \omega(\theta))$ and reward function $\hat{R}_g(s, a, \omega(\theta))$. Upon encountering a new task, the agent first estimates the hidden parameter and then feeds it into the learned transition function $\hat{T}_g$ and reward function $\hat{R}_g$. The agent will use them to generate samples and train a task-specific policy. The algorithm framework can be found in Algorithm 2 and Figure 1. Note that a similar paradigm has been widely used (Lee et al., 2020; Mendonca et al., 2020) in complex continuous control tasks.

We first derive a bound of the compounding error of dynamics prediction given the estimation error $\epsilon$ in hidden parameter at test time, with the Lipschitz model class (Asadi et al., 2018) assumption (we include a definition in appendix E):

**Theorem 4.1.** *Assume the estimation error for hidden parameter $\theta$ is bounded by $\epsilon$, that is, $W(\omega(\theta), \hat{\omega}(\theta)) \leq \epsilon$. Further assume an accurately learned $\hat{T}_g$ induced by a Lipschitz model class $F_g$ with the Lipschitz constant $K_{\hat{T}}^{\Omega}$ and $K_{\hat{T}}^S$, a fixed sequence of actions $a_0, \cdots, a_{n-1}$, and an initial state distribution $\mu(s)$. Then $\forall n \geq 1$, the $n$-step prediction error $\xi(n)$:*

$$\xi(n) := W(\hat{T}_g^n(\cdot|\mu, \omega), \hat{T}_g^n(\cdot|\mu, \hat{\omega})) \leq K_{\hat{T}}^{\Omega} \epsilon \sum_{i=0}^{n-1} (K_{\hat{T}}^S)^i, \tag{4}$$

*where $\hat{T}_g^n(\cdot|\mu, \omega)) := \hat{T}_g(\cdot|\hat{T}_g(\cdot|...\hat{T}_g(\cdot|\mu, a_0, \omega)..., a_{n-2}, \omega), a_{n-1}, \omega)$ is a generalized $n$-step transition function, and $\hat{T}_g^n(\cdot|\mu, \hat{\omega}))$ is defined similarly.*

The result shows how the multi-step prediction error $\xi$ of a generalized transition function in HiP-MDP scales with respect to the hidden parameter estimation error $\epsilon$. Additionally, the smoothness of the transition function also affects the multi-step prediction error. As we show in the experiments, this multi-step prediction bound can have a huge impact on the **planning** performance for model transfer.

Instead of directly planning, some methods choose to further learn the policy using the learned model. We investigate the value estimation difference and regret of model transfer induced by the hidden parameter estimation error. Note that the remaining theoretical results in this paper do not involve the Lipschitz model class assumption.

**Lemma 4.2.** *Given a HiP-MDP with learned $(K_{\hat{R}}^{\Omega}, K_{\hat{T}}^{\Omega})$-Lipschitz transition model $\hat{T}$, reward function $\hat{R}$, in **model transfer**, the generalized value function is Lipschitz continuous with respect to $\Omega$ with a constant bounded by:*

$$K_{V^{\pi}}^{\Omega} \leq K_{\hat{R}}^{\Omega} + \gamma K_{Q^{\pi}}^{\Omega} + \gamma K_{Q^{\pi}}^S K_{\hat{T}}^{\Omega}.$$

We get the above bound for the Lipschitz value function when assuming the Lipschitz continuity of transition and reward function with respect to only the hidden parameter. The lemma shows the relationship between $K_{V^{\pi}}^{\Omega}$ and $K_{Q^{\pi}}^{\Omega}$ in model transfer case. Then, by further assuming the Lipschitz continuity of dynamics & policy in Lemma 9, we can get the following bound leveraging the dual form of Wasserstein Metric (Equation 2), and computing the fixed point of recurrence.

**Theorem 4.3.** *Given a HiP-MDP with learned $(K_{\hat{R}}^S, K_{\hat{T}}^S, K_{\hat{R}}^A, K_{\hat{T}}^A, K_{\hat{R}}^{\Omega}, K_{\hat{T}}^{\Omega})$-Lipschitz transition dynamics $\hat{T}$, reward function $\hat{R}$, in **model transfer**, if $\gamma(K_T^S + K_{\pi}^S K_T^A) < 1$, the generalized value function corresponding to the $K_{\pi}^S$-Lipschitz policy $\pi(a|s)$ is Lipschitz continuous with respect to $\Omega$ with a constant bounded by:*

$$K_{V^{\pi}}^{\Omega} \leq \frac{K_R^{\Omega} + \gamma K_{Q^{\pi}}^S K_T^{\Omega}}{1 - \gamma}. \tag{5}$$

Now if we further use the bounds derived in Theorem 3.1 to substitute $K_{Q^{\pi}}^S$ (for Theorem 5.2 we substitute $K_{Q^{\pi}}^A$), we can get a bound of $K_{V^{\pi}}^{\Omega}$ that does not depend on $K_{Q^{\pi}}$.

Compared to Lemma 4.2, the bound derived in Theorem 4.3 does not depend on $K_{Q^{\pi}}^{\Omega}$ by leveraging Bellman equation (the details can be found in our proofs). In general, Theorem 4.3 characterizes

how the value estimation difference in model transfer is affected by the reward/transition functions' robustness to the hidden parameter. Theorem 4.3 and 3.1 together show that the value function of a HiP-MDP for model transfer is Lipschitz continuous under certain conditions.

Then we can derive the regret of model transfer given hidden parameter estimation error $\epsilon$. Here we further assume the agent gets the **optimal** policy in the "simulated environment" created by the learned transition and reward function. We first introduce the following lemma:

**Lemma 4.4.** *Given a $(K_R^S, K_T^S, K_R^\Omega, K_T^\Omega)$-Lipschitz HiP-MDP, for the optimal policy $\pi^*$, $|V^{\pi^*}(s, \omega_1(\theta)) - V^{\pi^*}(s, \omega_2(\theta))| \leq \max_{a \in A} |Q^{\pi^*}(s, a, \omega_1(\theta)) - Q^{\pi^*}(s, a, \omega_2(\theta))|$.*

Given the optimal policy, this lemma shows the relationship between value estimation difference and Q-value estimation difference. Combining the results in Corollary 3.2, we have:

**Lemma 4.5.** *Given a HiP-MDP with learned $(K_{\hat{R}}^S, K_{\hat{T}}^S, K_{\hat{R}}^\Omega, K_{\hat{T}}^\Omega)$-Lipschitz transition model $\hat{T}$, reward function $\hat{R}$, the optimal policy $\hat{\pi}^*(a|s)$ for $\hat{T}$ and $\hat{R}$, starting from state distribution $\mu(s)$, if $\gamma K_{\hat{T}}^S < 1$, the regret of model transfer given hidden parameter estimation error $\epsilon$ is bounded by:*

$$|J_{\pi^*}^\mu - J_{\hat{\pi}^*}^\mu| \leq \frac{K_{\hat{R}}^\Omega}{1 - \gamma} \cdot \epsilon + \frac{\gamma K_{\hat{R}}^S K_{\hat{T}}^\Omega}{(1-\gamma)(1 - \gamma K_{\hat{T}}^S)} \cdot \epsilon, \tag{6}$$

Besides the reward and transition functions' smoothness with respect to the hidden parameter $\theta$, the derived bound also shows that the distance between the expected return of the learned policy $\hat{\pi}^*$ and the environment's true optimal policy increases with $K_{\hat{R}}^S$ and $K_{\hat{T}}^S$, given the hidden parameter estimation error $\epsilon$. In other words, the performance of model transfer algorithms decreases as the sensitivity of the dynamics changes in states increases, which aligns with the intuition that the model-based methods need to accurately predict state transitions and expected rewards given states.

Furthermore, for the theorems derived in this section, we show that (proofs included in the appendix):

**Claim 4.6.** *Given a linear and deterministic transition function, the bound derived in Theorem 4.1 is tight. Given a linear and deterministic transition function and reward function, Lemma 4.5 is tight.*

## 5 Performance bounds of Policy Transfer in HiP-MDPs

We now focus on the regret bound induced by policy transfer algorithms in HiP-MDPs. We assume the error bound of the hidden parameters is the same (i.e., $d(\omega(\theta), \hat{\omega}(\theta)) \leq \epsilon$), and that the agent has learnt an accurate joint policy $\pi(a|s, \omega(\theta))$ during training. Upon encountering a new task, the agent feeds the estimated hidden parameter into the learned policy instead of the model. The agent will then directly use the joint policy to interact with the environment. The algorithm framework can be found in Algorithm 1 and Figure 1. Similar policy transfer paradigm has also been widely used (Yao et al., 2018; Rakelly et al., 2019). We investigate how the estimation error for the hidden parameter will affect the performance of the joint policy on a new task and compare with the performance bounds derived above for model transfer cases.

**Lemma 5.1.** *Given a HiP-MDP with learned $K_\pi^\Omega$-Lipschitz joint policy $\pi(a|s, \omega(\theta))$, in **policy transfer**, the generalized value function is Lipschitz continuous with respect to $\Omega$ with a constant bounded by:*

$$K_{V^\pi}^\Omega \leq K_{Q^\pi}^\Omega + K_{Q^\pi}^A K_\pi^\Omega.$$

We obtain the above bound for the Lipschitz value function when assuming the Lipschitz continuity of the learned policy with respect to only the hidden parameter. The lemma shows the relationship between $K_{V^\pi}^\Omega$ and $K_{Q^\pi}^\Omega$ in the policy transfer case. Then, similar to model transfer, if we further assume the Lipschitz continuity of the dynamics and the policy and compute the fixed point of the recurrence of the Bellman update, we can get the following bound:

**Theorem 5.2.** *Given a $(K_R^A, K_T^A, K_R^S, K_T^S)$-Lipschitz HiP-MDP, in **policy transfer**, the generalized value function corresponding to the pretrained $(K_\pi^S, K_\pi^\Omega)$-Lipschitz joint policy $\pi(a|s, \omega(\theta))$, if $\gamma(K_T^S + K_\pi^S K_T^A) < 1$, is Lipschitz continuous with respect to $\Omega$ with a constant bounded by*

$$K_{V^\pi}^\Omega \leq \frac{K_\pi^\Omega K_{Q^\pi}^A}{1 - \gamma}. \tag{7}$$

Compared to Lemma 5.1, the bound derived in Theorem 5.2 does not depend on $K_{Q^\pi}^\Omega$ by leveraging Bellman equation. Theorem 5.2 characterizes how the value estimation difference in policy transfer is affected by the policy's robustness to the hidden parameter. Theorem 5.2 and 3.1 together show the value function of a HiP-MDP for policy transfer is Lipschitz continuous under certain conditions. Different from the bound in Theorem 4.3 for model transfer, for policy transfer methods this value estimation error is also affected by the policy's robustness with respect to the hidden parameters, as well as the Q-value's robustness to the **actions** (only with respect to **states** in Theorem 4.3), with no dependence on the reward and transition functions' smoothness with respect to the hidden parameters.

We can further derive the Lipschitz constant of the expected discounted reward (regret) starting from the state distribution $\mu$. Similar to what we did in model transfer case, if we also assume the learned joint policy over $\omega(\theta)$ is optimal, by leveraging the results in Corollary 3.2, we have:

**Lemma 5.3.** *Given a $(K_R^A, K_T^A, K_R^S, K_T^S)$-Lipschitz HiP-MDP, a pretrained $K_\pi^\Omega$-Lipschitz optimal joint policy $\pi(a|s, \omega(\theta))$, starting from state distribution $\mu(s)$, if $\gamma K_T^S \leq 1$, the regret of policy transfer induced by hidden parameter estimation error $\epsilon$ is bounded by:*

$$|J_{\pi^*}^\mu - J_\pi^\mu| \leq \frac{\gamma K_\pi^\Omega (K_R^A + \gamma K_R^S K_T^A - \gamma K_T^S K_R^A)}{(1-\gamma)(1-\gamma K_T^S)} \cdot \epsilon, \tag{8}$$

Comparing the derived performance bound with the one in Lemma 4.5, we find that the distance between the expected return of the learned joint policy $\pi$ and the environment's true optimal policy $\pi^*$ increases also with $K_T^A$ and $K_R^A$, besides $K_T^S$ and $K_R^S$. In other words, the performance of the policy transfer decreases as the sensitivity of the dynamics changes in both states and actions increases, while the model-based method is not quite sensitive to the dynamics change in actions. This result is reasonable as policy-based method is influenced mostly by direct impact of actions. Regarding the tightness of the derived bound, we get similar results as for model transfer:

**Claim 5.4.** *Given a linear and deterministic transition function, reward function and policy, the bounds derived in Lemma 5.3 are tight.*

**Summary** Besides characterizing the robustness of model and policy transfer methods with respect to hidden parameter estimation error separately, Lemma 4.5 and 5.3 also imply that, policy transfer methods are expected to perform more robust when difference in effects of different **actions** is small, whereas model transfer methods are expected to perform better when neighboring **states** have relatively similar dynamics. Given these bounds, one direct implication to HiP-MDP algorithms in practice is that we can infer the performance trend of policy transfer and model transfer algorithms by either qualitatively or quantitatively estimating the dynamics and reward functions' sensitivity to states and actions. This can further help us determine which method is probably more advantageous for a specific HiP-MDP problem. Note that $K_{\hat{T}}^\theta$ and $K_\pi^\Omega$ in general can be dependent on the model class (architecture) of the model (e.g. neural nets) used to parameterized the policy or model.

## 6    EMPIRICAL EVALUATION

We now corroborate our theoretical results in relatively large scenarios where neural networks are needed to parameterize models and polices. We evaluate model and policy transfer methods on two continuous scenarios, *ball-goal* and *ball-wind*, where we can quantitatively estimate the influence of changing environment hyperparameters on different Lipschitz constants. Then by investigating whether the empirical performance changing is consistent with what we expect from theoretical results, we can corroborate whether our derived theorems match what happened in practice.

For both scenarios, the goal of the agent is to control a ball to quickly reach a target position. In *ball-goal*, the angle of the goal direction is the hidden parameter. That is, the goal direction changes across tasks, and the agent will obtain maximum cumulative reward if it figures out the right angle for the current task and keeps moving in that direction until reaching the goal. In *ball-wind*, the goal position is fixed across different tasks but there is wind across the plane with different directions. The angle of the wind direction is the hidden parameter in this scenario. The reward function consists of a dense reward proportional to how closer the ball moves towards the goal compared to last step, a control cost, as well as a larger final reward for reaching the goal. Thus for *ball-goal*, only the reward function is changing with respect to different hidden parameters, while in *ball-wind*, both reward and transition functions are changing. More details about the environments can be found in Appendix B.

For each HiP-MDP, we manually calculate the value of different Lipschitz constants approximately given the actual reward and transition functions for both scenarios. We further calculate the bounds derived in Lemma 4.5 and Lemma 5.3 using the results. The Lipschitz constants are given in Table 1. We mainly investigate the effect of the step size $v$, goal distance $g$, and state accelerator $m$ (*ball-wind*). The results imply how different Lipschitz constants are changing as we change those environmental hyper-parameters if our derived bounds are relatively tight and close to what happened in practice.

| Env | $K_T^S$ | $K_T^A$ | $K_T^\Omega$ | $K_R^S$ | $K_R^A$ | $K_R^\Omega$ | $K_\pi^\Omega$ | $K_J$ (Model transfer) | $K_J$ (Policy transfer) |
|---|---|---|---|---|---|---|---|---|---|
| *ball-goal* | 1 | $v$ | 0 | 2 | $v$ | $2g$ | $g$ | $\frac{2g}{1-\gamma}$ | $\frac{\gamma(1+\gamma)vg}{(1-\gamma)^2}$ |
| *ball-wind* | $m$ | 1 | $v$ | $m+1$ | 1 | $v$ | $\frac{\sqrt{2}v}{\sqrt{2}v-m(g-1)+g}$ | $\frac{v+\gamma}{(1-\gamma)(1-\gamma m)}$ | $K_\pi^\Omega \cdot \frac{\gamma(1+\gamma)}{(1-\gamma m)(1-\gamma)}$ |

Table 1: Lipschitz constants for *ball-goal* and *ball-wind*

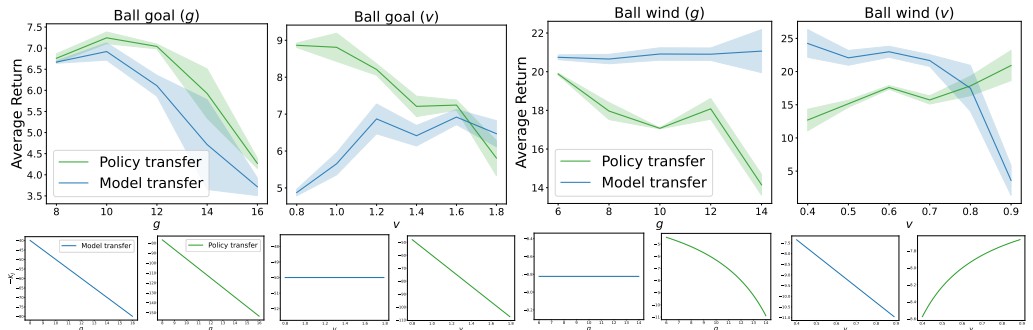

Figure 2: First row: Average return on test tasks v.s. environmental properties ($v$ and $g$). Second row: $-K_J$ v.s. $v$ and $g$: how $-K_J$ is expected to change based on our our theoretical computing results in Table 1. Each plot in the first row is associated with two plots below with corresponding color. The performance of the algorithm is expected to decrease if $-K_J$ decreases.

## 6.1 EMPIRICAL RESULTS CORRESPONDING TO LEMMA 4.5 AND LEMMA 5.3

As both Lemma 4.5 and Lemma 5.3 can be written in the form $|J_{\hat{\pi}} - J_{\pi^*}| < K_J \cdot \epsilon$, we investigate how $K_J$ is changing as we change different environmental hyperparameters both **theoretically**: manually computing the Lipschitz constants given the environment's dynamics (appendix B) and the two lemmas (results shown in Table 1), and **empirically**: investigating how the algorithm's empirical performance is changing as we alter the environment's hyperparameters ($K_J$ is expected to inversely related to the expected return), and see whether the theoretical and empirical results are consistent.

For empirical training, we use state-of-the-art meta-RL algorithms PEARL (Rakelly et al., 2019) as the policy transfer method to learn the joint policy, as well as CaDM (Lee et al., 2020) to learn the dynamics model. The context encoder (probabilistic encoder) is updated together with loss from Q-value prediction like in PEARL, as well as loss from dynamics model prediction like in CaDM. Given a new task, we let the agent collect transitions by interacting with the environment for only one episode with randomly initialized hidden-parameter distribution.[2] Then the agent will use the learned context encoder to infer a distribution over the hidden parameter of the current task. This distribution over the hidden parameter is fixed afterwards for the current task, i.e. $\epsilon$ is fixed and the same for both methods. Then, for policy transfer, we sample from the hidden parameter distribution and directly feed that inferred latent parameter into the learned joint policy. We evaluate the average return of this policy on the current task. For model transfer, we feed that inferred latent parameter into the learned transition model and reward function. We use the models to generate samples and run SAC (Haarnoja et al., 2018) to learn the task-specific policy for this "simulated environment".

The empirical results are shown in Figure 2. We plot $-K_J$ v.s. $v/g$ using the results in Table 1. Recall that $K_J$ is expected to **inversely** related to the average return. i.e., **increase** in $-K_J$ implies

---

[2]The hidden parameters here and the inferred hidden parameters mentioned below are all referring to the mapping of the hidden parameter in the latent representation space through the context encoder. We assume the true distribution over the hidden parameters is also implicitly mapped into the latent space.

a decrease in the regret of the algorithm, thus the average return is expected to **increase** if our theoretical results match what happened in practice. Almost all of the results show that the change of the expected return is inversely related to the change of the approximated Lipschitz constant $K_J$. The environment parameters directly affect $K_J$ and the trend of the performance change in a way that is highly consistent with our theoretical analysis. In general, we find that **a slower increase in Lipschitz constant with respect to the hidden parameter implies a smaller performance decay**. The empirical results implies that for real-world HiP-MDP tasks, if we can quantitatively estimate the Lipschitz constant $K_J$ for model and policy transfer methods given the environment parameters, we can estimate which one is expected to have better final performance. For instance, in ball-wind, if $g$ is quite large, given the computed $K_J$, we will expect model transfer to have better performance as it is less affected by the value of $g$ compared to policy transfer method. Note that in our scenarios, the performance drop is mainly induced by the estimation error of the hidden parameters, and the learned dynamics and policies are near optimal. We show the empirical evidence in appendix C.

## 6.2 EMPIRICAL RESULTS CORRESPONDING TO THEOREM 4.1

We also show how the bound of the multi-step prediction error derived in Theorem 4.1 is related to the performance of the other planning method used in model transfer — Model Predictive Control (MPC) (Garcia et al., 1989). MPC is widely used in recent deep model-based RL works (Chua et al., 2018; Nagabandi et al., 2019). Different from the previous approaches, MPC does not need to explicitly estimate the value function and its performance is directly affected by the multi-step prediction error. With the results in Table 1, we can get the multi-step prediction upper bound for *ball-wind* using Theorem 4.1: $\xi(n) \leq v \sum_{i=0}^{n-1} m^i \cdot \epsilon$. The bound

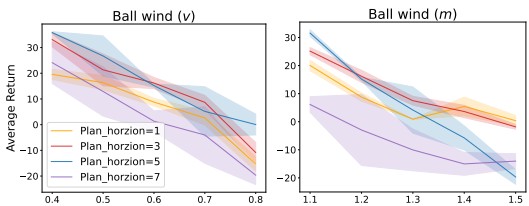

Figure 3: Average return on test tasks v.s. environmental properties ($m$ & $v$) that affect the Lipschitz constants for MPC method on *ball-wind*.

implies that if the theorem matches what happens in practice, the $n$-step prediction error will increase as we increase $v$ and $m$, and thus **the performance is expected to drop as we increase $v$ and $m$**. (The performance of MPC and the multi-step prediction error are inversely correlated.) Similar to the previous subsection, we investigate whether the empirical performance of MPC matches our predictions from theoretical results.

During empirical evaluation, we first let the agent infer the hidden parameter and feed that into the learned transition function and reward function. Then at each time step, we let the agent randomly sample a large number of actions, and use the learned model to predict $N$ steps into the future for each of them and choose the action with the highest predicted cumulative reward. We investigate how $v$ and $m$ affect the algorithm's final performance empirically. The results are shown in Figure 3, the average return of the algorithm on test tasks decreases as we increase the value of $v$ and $m$, which is consistent with our theoretical result. We also find that the variance of the performance (the width of error bar) is increasing as we use longer planning horizon for MPC.

## 7 CONCLUSION

Assuming the learned model/policy is optimal, we investgiated how the hidden parameter estimation error affects the robustness of model and policy transfer algorithms from a theoretical perspective, respectively. We show the conditions under which the value functions of HiP-MDPs are Lipschitz continuous. We further derive regret bounds for model and policy transfer, which are proved to be tight in linear and deterministic cases. Our empirical results are consistent with the theoretical results and indicate that a faster increase in Lipschitz constant with respect to the hidden parameter implies a larger performance decay. We note that in real-world HiP-MDP problems, especially when applying deep RL algorithms, the suboptimality of the learned model/policy can still play an important role in the potential performance drop on a new task, independently of the estimation error of the hidden parameter. And in many cases, policy transfer performs better than model transfer simply because the downstream model-free deep RL algorithms are consistently better in regular MDPs.

## 8 ACKNOWLEDGEMENT

The authors would like to thank Kavosh Asadi, Saket Tiwari, Michael Littman for discussions and helpful feedback, and the anonymous reviewers for valuable feedback that improved the paper substantially. This work was supported in part by an NSF Graduate Research Fellowship under grant #2040433, NSF grants #1717569 #1955361 #IIS-2007076 and CAREER award #1844960, DARPA grant W911NF1820268, and ONR contracts N00014-17-1-2699, and was conducted using computational resources and services at the Center for Computation and Visualization, Brown University. The U.S. Government is authorized to reproduce and distribute reprints for Governmental purposes notwithstanding any copyright notation thereon. The content is solely the responsibility of the authors and does not necessarily represent the official views of DARPA, the NSF, the ONR, or the AFOSR.

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

## A   MODEL TRANSFER AND POLICY TRANSFER

---

**Algorithm 1:** Policy transfer

---

**Input:** Test tasks $\{\tau_i\}_{i=1:K}$ with hidden parameters $\theta_i \sim P_\Omega$
**for** each task $\tau_i$ **do**
    Roll out policy $\pi(a|s, \tilde{\omega}(\theta))$ with randomly generated Gaussian $\tilde{\omega}(\theta)$ to collect data
    $c = \{s_1, a_1, r_1, \cdots, s_n, a_n, r_n\}$
    Infer the distribution $\hat{\omega}(\theta)$ over the hidden parameter of task $\tau_i$ with the learned encoder
    $q(\omega(\theta)|c)$
    Roll out policy $\pi(a|s, \hat{\omega}(\theta))$ to interact with environment (**evaluation**)
**end for**

---

**Algorithm 2:** Model transfer

---

**Input:** Test tasks $\{\tau_i\}_{i=1:K}$ with hidden parameters $\theta_i \sim P_\Omega$
**for** each task $\tau_i$ **do**
    Roll out policy $\pi(a|s, \tilde{\omega}(\theta))$ with randomly generated Gaussian $\tilde{\omega}(\theta)$ to collect data
    $c = \{s_1, a_1, r_1, \cdots, s_n, a_n, r_n\}$
    Infer the distribution $\hat{\omega}(\theta)$ over the hidden parameter of task $\tau_i$ with the learned encoder
    $q(\omega(\theta)|c)$
    Planning with learned transition function $T_g(s'|s, a, \hat{\omega}(\theta))$ and reward function $R_g(s, a, , \hat{\omega}(\theta))$
    (**evaluation**)
**end for**

---

Empirically, in typical deep meta-RL settings (Finn et al., 2017; Rakelly et al., 2019), the true value of hidden parameters are not known to the agent both during training phase and evaluation phase. We follow this setting in out experiments and train a context encoder like the previous approaches to infer the hidden parameter in **latent representation space**. Then the true distribution over hidden parameter is also assumed to be implicitly mapped into the latent representation space. Our empirical experiments show that our theories can be extended to hidden parameter estimation in the latent space, and the results show consistent patterns with the theoretical results.

## B   EXPERIMENT SETTINGS

**Ball-goal**: The state space consists of the ball's $x, y$ coordinates. The action space consists of the ball's $x, y$ velocity. $v$ controls the step size in the direction of $x$ axis in the transition function. The reward is proportional to the how far the agent has moved towards the goal's position $(g\cos\phi_g, g\sin\phi_g)$. We fix the value of $v, g$ and use $\phi_g$ as the hidden parameter to create HiP-MDPs. Details are shown below:

Figure 4: The Ball environment used in our experiments.

- State space: $\{s_x, s_y\}$
- Action space: $\{a_x, a_y\}$
- Transition function:

$$s'_x = s_x + a_x \cdot v$$
$$s'_y = s_y + a_x$$

- Reward function:

$$R = \sqrt{(s_x - g\cos\phi_g)^2 + (s_y - g\sin\phi_g)^2} - \sqrt{(s'_x - g\cos\phi_g)^2 + (s'_y - g\sin\phi_g)^2}$$

**Ball-wind**:The state space consists of the ball's $x, y$ coordinates. The action (one dimension) describes the direction of the agent's next move. In the transition function, $m$ is describes the value of state accelerator, $v$ controls the step size of the wind, $\theta$ describes the direction of the wind. The reward is

proportional to the how far the agent has moved towards the goal's position $(g\cos\phi_g, g\sin\phi_g)$, plus a goal reward when succeeds and a control penalty $(-0.5)$. We fix the value of $v, g, m, \phi_g$ and use $\theta$ as the hidden parameter to create HiP-MDPs. Details are shown below:

- State space: $\{s_x, s_y\}$

- Action space: $\{a\}$

- Transition function:

$$s'_x = ms_x + \cos a - v\cos\theta$$
$$s'_y = ms_y + \sin a - v\sin\theta$$

- Reward function:

$$R = \sqrt{(s_x - g\cos\phi_g)^2 + (s_y - g\sin\phi_g)^2} - \sqrt{(s'_x - g\cos\phi_g)^2 + (s'_y - g\sin\phi_g)^2}$$
$$+ \mathbb{1}\{\sqrt{(s'_x - g\cos\phi_g)^2 + (s'_y - g\sin\phi_g)^2} \le 0.1\} \cdot 20 - 0.5$$

When computing the Lipschitz constants for Ball-wind, we only consider the difference of distance towards goal part in the reward function for ease of calculation.

## C  ADDITIONAL EXPERIMENTS WHEN THE HIDDEN PARAMETER IS BETTER INFERRED

We further show that in these two scenarios, the performance drop as we change environment parameters is mainly induced by the estimation error of the hidden parameters, and the learned transition/reward functions and policies are near optimal. As shown in Figure 5, when we let the agent interact with the environments for more trajectories and keep using the collected data to infer the hidden parameter, changing of the environment's Lipschitz constant does not affect the performance as much as before when the estimation of the hidden parameter is far less accurate.

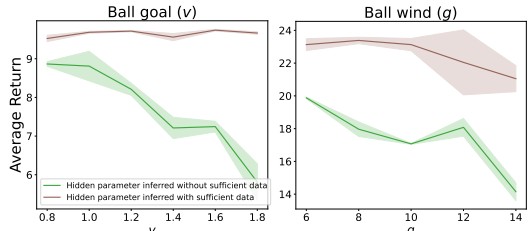

Figure 5: Average return on test tasks v.s. environmental properties ($v$ & $g$) for different estimation accuracy of the hidden parameter in *ball-goal* & *ball-wind*.

## D  COMPARISON OF THEOREM 3.1 WITH PREVIOUS RESULTS

**Assumption 1.** The policy is optimal.

**Assumption 2.** $S$ and $A$ share the same Lipschitz constant.

| Theorems | Value difference bound | Assumption 1 | Assumption 2 |
|---|---|---|---|
| **Theorem 3.1** | $\frac{K_R^S + K_R^A K_\pi^S}{1 - \gamma(K_T^S + K_T^A K_\pi^S)}$ | ✗ | ✗ |
| (Asadi et al., 2018; Gelada et al., 2019) | $\frac{K_R^S}{1 - \gamma K_T^S}$ | ✓ | ✗ |
| (Rachelson & Lagoudakis, 2010; Pirotta et al., 2015) | $\frac{K_R^S(1 + K_\pi^S)}{1 - \gamma K_T^S(1 + K_\pi^S)}$ | ✗ | ✓ |

Table 2: Comparison of Theorem 3.1 with previous results in regular MDPs

Theorem 3.1 and Corollary 3.2 also directly apply to a $(K_R^S, K_T^S, K_R^A, K_T^A)$-Lipschitz **regular** MDP with a $K_\pi^S$-Lipschitz policy, as the Lipschitz continuity assumptions with respect to hidden parameters are not used in both the final results and proofs. In this sense, we show a comparison of Theorem 3.1

with previous results in regular MDPs in Table 2. Note that in Theorem 3.1, we have:

$$K_{V^\pi}^S \le K_{Q^\pi}^S + K_{Q^\pi}^A K_\pi^S \le \frac{K_R^S + K_R^A K_\pi^S}{1 - \gamma(K_T^S + K_T^A K_\pi^S)}$$

Specifically, the upper bound for $K_{Q^*}^S$ in Corollary 3.2 recovers the results in (Asadi et al., 2018; Gelada et al., 2019). If we assume the Lipschitz constants for state and action are the same, we get Corollary E.1, which recovers the results in (Rachelson & Lagoudakis, 2010; Pirotta et al., 2015). For **regular** MDPs, our results are more general compared to prior works as (1) we do not assume that the Lipschitz constants for state and action are the same because this is usually not the case in practice; (2) our theory applies to all Lipschitz policies rather than only the optimal policy.

## E  Proofs and More Theory Results

Using the generalized functions defined in Section 2, we can easily extend Bellman update rule to HiP-MDPs:

$$Q_{n+1}(s, a, \omega(\theta)) \leftarrow R_g(s, a, \omega(\theta)) + \gamma \int \hat{T}_g(s'|s, a, \omega(\theta)) V_n^\pi(s', \omega(\theta)) ds', \tag{9}$$

where $Q_{n+1}$ converges to $Q^*$ as $n \to \infty$.

**Theorem 3.1**. For a $(K_R^S, K_T^S, K_R^A, K_T^A, K_R^\Omega, K_T^\Omega)$-Lipschitz HiP-MDP with a $(K_\pi^S, K_\pi^\Omega)$-Lipschitz policy $\pi$, if $\gamma(K_T^S + K_\pi^S K_T^A) < 1$, the value function is Lipschitz continuous with respect to $S\&A$ with constants bounded by:

$$K_{Q^\pi}^A := \sup_\omega \sup_s \sup_{a_1, a_2} \frac{|Q(s, a_1, \omega) - Q(s, a_2, \omega)|}{d_A(a_1, a_2)} \le \frac{K_R^A - \gamma(K_R^A K_T^S - K_R^S K_T^A)}{1 - \gamma(K_T^S + K_\pi^S K_T^A)}$$

$$K_{Q^\pi}^S := \sup_\omega \sup_a \sup_{s_1, s_2} \frac{|Q(s_1, a, \omega) - Q(s_2, a, \omega)|}{d_S(s_1, s_2)} \le \frac{K_R^S - \gamma K_\pi^S(K_R^S K_T^A - K_R^A K_T^S)}{1 - \gamma(K_T^S + K_\pi^S K_T^A)}$$

$$K_{V^\pi}^S := \sup_\omega \sup_a \sup_{s_1, s_2} \frac{|V(s_1, \omega) - V(s_2, \omega)|}{d_S(s_1, s_2)} \le K_{Q^\pi}^S + K_{Q^\pi}^A K_\pi^S.$$

*Proof.* The proof is mainly based on the Bellman update rule (9), the dual form of Wasserstein Metric (Equation 2), and fixed point iteration.

Recall that:

$$Q_{n+1}(s, a, \omega(\theta)) \leftarrow R_g(s, a, \omega(\theta)) + \gamma \int \hat{T}_g(s'|s, a, \omega(\theta)) V_n^\pi(s', \omega(\theta)) ds'$$

Now let:

$$K_{Q^\pi, n+1}^A := \sup_\omega \sup_s \sup_{a_1, a_2} \frac{|Q_{n+1}(s, a_1, \omega) - Q_{n+1}(s, a_2, \omega)|}{d_A(a_1, a_2)} \le \sup_\omega \sup_s \sup_{a_1, a_2} \frac{|R(s, a_1, \omega) - R(s, a_2, \omega)|}{d_A(a_1, a_2)} +$$

$$\gamma \sup_\omega \sup_s \sup_{a_1, a_2} \frac{\left| \int_{s'} (T(s'|s, a_1, \omega) - T(s'|s, a_2, \omega)) V^\pi(s', \omega) ds' \right|}{d_A(a_1, a_2)}$$

$$= K_R^A + \gamma \sup_\omega \sup_s \sup_{a_1, a_2} \frac{K_{V^\pi}^S \left| \int_{s'} (T(s'|s, a_1, \omega) - T(s'|s, a_2, \omega)) \frac{V^\pi(s', \omega)}{K_{V^\pi}^S} ds' \right|}{d_A(a_1, a_2)}$$

$$\le K_R^A + \gamma \sup_\omega \sup_{f: K_f^S \le 1} \sup_s \sup_{a_1, a_2} \frac{K_{V^\pi}^S \left| \int_{s'} (T(s'|s, a_1, \omega) - T(s'|s, a_2, \omega)) f(s', \omega) ds' \right|}{d_A(a_1, a_2)}$$

$$\le K_R^A + \gamma K_{V^\pi, n}^S K_T^A$$

$$\tag{10}$$

The last inequality holds according to Equation 2.

Similarly, we can get:

$$K_{Q^\pi}^S := \sup_\omega \sup_a \sup_{s_1,s_2} \frac{|Q_{n+1}(s_1,a,\omega) - Q_{n+1}(s_2,a,\omega)|}{d_S(s_1,s_2)} \leq \sup_\omega \sup_a \sup_{s_1,s_2} \frac{|R(s_1,a,\omega) - R(s_2,a,\omega)|}{d_A(a_1,a_2)} +$$

$$\gamma \sup_\omega \sup_a \sup_{s_1,s_2} \frac{\left| \int_{s'} (T(s'|s_1,a,\omega) - T(s'|s_2,a,\omega))V^\pi(s',\omega)ds' \right|}{d_S(s_1,s_2)}$$

$$= K_R^S + \gamma \sup_\omega \sup_a \sup_{s_1,s_2} \frac{K_{V^\pi}^S \left| \int_{s'} (T(s'|s_1,a,\omega) - T(s'|s_2,a,\omega))\frac{V^\pi(s',\omega)}{K_{V^\pi}^S}ds' \right|}{d_S(s_1,s_2)}$$

$$\leq K_R^S + \gamma \sup_\omega \sup_{f:K_f^S \leq 1} \sup_a \sup_{s_1,s_2} \frac{K_{V^\pi}^S \left| \int_{s'} (T(s'|s_1,a,\omega) - T(s'|s_2,a,\omega))f(s',\omega)ds' \right|}{d_S(s_1,s_2)}$$

$$\leq K_R^S + \gamma K_{V^\pi,n} K_T^S \tag{11}$$

We need to further derive the upper bound of $K_V$:

$$K_{V^\pi}^S := \sup_\omega \sup_{s_1,s_2} \frac{|V(s_1,\omega) - V(s_2,\omega)|}{d_S(s_1,s_2)} = \sup_\omega \sup_{s_1,s_2} \frac{|Q(s_1,\pi(s_1),\omega) - Q(s_2,\pi(s_2),\omega)|}{d_S(s_1,s_2)}$$

$$\leq \sup_\omega \sup_{s_1,s_2} \frac{|Q(s_1,\pi(s_1),\omega) - Q(s_1,\pi(s_2),\omega)|}{d_S(s_1,s_2)} + \sup_\omega \sup_{s_1,s_2} \frac{|Q(s_1,\pi(s_2),\omega) - Q(s_2,\pi(s_2),\omega)|}{d_S(s_1,s_2)}$$

$$\leq K_Q^A K_\pi^S + K_Q^S \tag{12}$$

Plugging Eqn 12 into Eqn 10 & Eqn 11, we get:

$$K_{Q,n+1}^A \leq K_R^A + \gamma K_T^A (K_{Q,n}^S + K_{Q,n}^A K_\pi^S) \tag{13}$$

$$K_{Q,n+1}^S \leq K_R^S + \gamma K_T^S (K_{Q,n}^S + K_{Q,n}^A K_\pi^S) \tag{14}$$

By computing the fixed point of the recurrence, we get:

$$K_{Q^\pi}^A = \lim_{n\to\infty} K_{Q,n+1}^A \leq \frac{K_R^A - \gamma(K_R^A K_T^S - K_R^S K_T^A)}{1 - \gamma(K_T^S + K_\pi^S K_T^A)}, \tag{15}$$

$$K_{Q^\pi}^S = \lim_{n\to\infty} K_{Q,n+1}^S \leq \frac{K_R^S - \gamma K_\pi^S(K_R^S K_T^A - K_R^A K_T^S)}{1 - \gamma(K_T^S + K_\pi^S K_T^A)}. \tag{16}$$

□

**Corollary E.1.** *If we assume the Lipschitz constants for state and action are the same (* $K_R^A = K_R^S = K_R^{S,A}, K_T^A = K_T^S = K_R^{S,A} = K_T^{S,A}$*):*

$$|R_g(s_1,a_1,\omega) - R_g(s_2,a_2,\omega)| \leq K_R^{S,A} d_{S,A}((s_1,a_1),(s_2,a_2))$$

$$W(T_g(\cdot|s_1,a_1,\omega), T_g(\cdot|s_2,a_2,\omega)) \leq K_T^{S,A} d_{s,a},$$

*then:*

$$K_{Q^\pi}^{S,A} = K_{Q^\pi}^S = K_{Q^\pi}^A \leq \frac{K_R^{S,A}}{1 - \gamma K_T^{S,A}(1 + K_\pi^S)}$$

*Proof.* Let $K_R^A = K_R^S = K_R^{S,A}, K_T^A = K_T^S = K_R^{S,A} = K_T^{S,A}$ in the bounds of Theorem 3.1. □

**Corollary 3.2.** If the policy $\pi$ is optimal and $\gamma K_T^S < 1$, then:

$$K_{Q^*}^S \leq \frac{K_R^S}{1 - \gamma K_T^S}, \qquad K_{Q^*}^A \leq \frac{K_R^A + \gamma K_R^S K_T^A - \gamma K_T^S K_R^A}{1 - \gamma K_T^S}.$$

*Proof.* When the policy is optimal, we have (Asadi et al., 2018; Gelada et al., 2019):

$$|V^{\pi^*}(s_1, \omega) - V^{\pi^*}(s_2, \omega)| = |\max_{a \in A} Q^{\pi^*}(s_1, a, \omega) - \max_{a \in A} Q^{\pi^*}(s_2, a, \omega)|$$

$$\leq \max_{a \in A} |Q^{\pi^*}(s_1, a, \omega) - Q^{\pi^*}(s_2, a, \omega)|$$

Thus,

$$K_V^S \leq K_Q^S = \sup_{\omega} \sup_{a} \sup_{s_1, s_2} \frac{|Q(s_1, a, \omega) - Q(s_2, a, \omega)|}{d_S(s_1, s_2)} \tag{17}$$

Plugging Eqn 17 into Eqn 10 and Eqn 11, we get:

$$K_{Q,n+1}^A \leq K_R^A + \gamma K_{Q,n}^S K_T^A \tag{18}$$

$$K_{Q,n+1}^S \leq K_R^S + \gamma K_{Q,n}^S K_T^S \tag{19}$$

First compute the fixed point of the recurrence of Eqn 19, we get:

$$K_{Q^*}^S = \lim_{n \to \infty} K_{Q,n+1}^S \leq \frac{K_R^S}{1 - \gamma K_T^S} \tag{20}$$

Now plugging Eqn 20 back into Eqn 18:

$$K_{Q^*}^A = \lim_{n \to \infty} K_{Q,n+1}^A \leq K_R^A + \gamma \frac{K_R^S}{1 - \gamma K_T^S} K_T^A = \frac{K_R^A + \gamma K_R^S K_T^A - \gamma K_T^S K_R^A}{1 - \gamma K_T^S} \tag{21}$$

$\square$

**Definition E.2.** Given a metric state space $(S, d_S)$, an action space $A$, and a metric hidden-parameter space $(\Theta, d_\Theta)$, we define $F_g$ as a collection of functions: $F_g = \{f : S \times \Theta \mapsto S\}$ distributed according to $g(f|a)$ where $a \in A$. We say that $(F_g^S, F_g^\Theta)$ is a Lipschitz model class in HiP-MDPs if

$$K_T^S := \sup_{f \in F_g^S} K_f^{S,S}, \tag{22}$$

and

$$K_T^\Theta := \sup_{f \in F_g^\Theta} K_f^{\Theta,S} \tag{23}$$

are finite.

The transition function associated with a Lipschitz model class then can be defined by:

$$T(s'|s, a, \theta) = \sum_f \mathbb{1}(f(s, \theta) = s') g(f|a), \tag{24}$$

And the generalized transition function associated with a Lipschitz model class would be:

$$T_g(s'|s, a, \omega) = \int_\theta \sum_f \mathbb{1}(f(s, \theta) = s') g(f|a) \omega(\theta) d\theta \tag{25}$$

We introduce the following two lemmas regarding Lipschitz model class (Asadi et al., 2018)(We also include a definition in appendix):

**Lemma E.3.** *(Asadi et al., 2018) A generalized transition function $T_g$ induced by a Lipschitz model class $F_g$ and fixed $(\theta, a)$ is Lipschitz with a constant:*

$$K_T^\mu = K_{W,W}^{A,\Theta}(T_g) := \sup_{\theta \in \Theta} \sup_{a \in A} \sup_{\mu_1, \mu_2} \frac{W(T_g(\cdot|\mu_1, a, \theta), T_g(\cdot|\mu_2, a, \theta))}{W(\mu_1, \mu_2)} \leq K_T^S \tag{26}$$

**Lemma E.4.** *A generalized transition function $T_g$ induced by a Lipschitz model class $F_g$ and fixed $(s, a)$ is Lipschitz with a constant:*

$$K_T^\Omega = K_{W,W}^{A,S}(T_g) := \sup_{s \in S} \sup_{a \in A} \sup_{\omega_1, \omega_2} \frac{W(T_g(\cdot|s, a, \omega_1), T_g(\cdot|s, a, \omega_2))}{W(\omega_1, \omega_2)} \leq K_T^\Theta \tag{27}$$

Intuitively, the above Lemmas give a bound on the differences in distributions an agent transitions to, as a function of how different the distributions of states it transitions from are, or how different the distributions of the hidden parameters are. It is theoretically pleasing that these bounds are given as the Lipschitz constants for the corresponding differences between point distributions of either states or hidden parameters.

Proof for **Lemma E.4**

*Proof.*

$$
\begin{aligned}
W(T_g(\cdot|s,a,\omega_1), T_g(\cdot|s,a,\omega_2)) &= \sup_{h:K_{d_S,\mathbb{R}}(h)\leq 1} \int_{s'} (T_g(s'|s,a,\omega_1) - T_g(s'|s,a,\omega_2))h(s')ds' \\
&= \sup_{h:K_{d_S,\mathbb{R}}(h)\leq 1} \int_{s'} \int_{\theta} T(s'|s_0,a,\theta)(\omega_1(\theta) - \omega_2(\theta))h(s')ds'd\theta \\
&= \sup_{h:K_{d_S,\mathbb{R}}(h)\leq 1} \int_{s'} \int_{\theta} \sum_t g(t|a)\mathbb{1}(t(s_0,\theta)=s')(\omega_1(\theta) - \omega_2(\theta))h(s')ds'd\theta \\
&= \sup_{h:K_{d_S,\mathbb{R}}(h)\leq 1} \sum_t g(t|a) \int_{\theta} \int_{s'} \mathbb{1}(t(s_0,\theta)=s')(\omega_1(\theta) - \omega_2(\theta))h(s')ds'd\theta \\
&= \sup_{h:K_{d_S,\mathbb{R}}(h)\leq 1} \sum_t g(t|a) \int_{\theta} (\omega_1(\theta) - \omega_2(\theta))h(t(s_o,\theta))d\theta \\
&\leq \sum_t g(t|a) \sup_{h:K_{d_S,\mathbb{R}}(h)\leq 1} \int_{\theta} (\omega_1(\theta) - \omega_2(\theta))h(t(s_o,\theta))d\theta \\
&= K_T^\Theta \sum_t g(t|a) \sup_{h:K_{d_S,\mathbb{R}}(h)\leq 1} \int_{\theta} (\omega_1(\theta) - \omega_2(\theta))\frac{h(t(s_o,\theta))}{K_T^\Theta}d\theta \\
&\leq K_T^\Theta \sum_t g(t|a) \sup_{q:K_{d_S,\mathbb{R}}(q)\leq 1} \int_{\theta} (\omega_1(\theta) - \omega_2(\theta))q(s_o,\theta)d\theta \\
&= K_T^\Theta \sum_t g(t|a)W(\omega_1,\omega_2) = K_T^\Theta W(\omega_1,\omega_2)
\end{aligned}
\tag{28}
$$

$\square$

Using Lemma E.3, we can derive the bound of the compounding error of dynamics prediction given the estimation error $\epsilon$ in hidden parameter at test time (Theorem 4.1).

**Theorem 4.1**. Assuming the estimation for hidden parameter $\theta$ is bounded by $\epsilon$, that is, $W(\omega(\theta), \hat{\omega}(\theta)) \leq \epsilon$, an learned accurate $\hat{T}_g$ induced by a Lipschitz model class $F_g$ with the Lipschitz constant $K_{\hat{T}}^\Omega$ and $K_{\hat{T}}^S$, a fixed sequence of actions $a_0, \cdots, a_{n-1}$, and a start state distribution $\mu$. Then $\forall n \geq 1$:

$$
\xi(n) := W(\hat{T}_g^n(\cdot|\mu,\omega), \hat{T}_g^n(\cdot|\mu,\hat{\omega})) \leq K_{\hat{T}}^\Omega \epsilon \sum_{i=0}^{n-1} (K_{\hat{T}}^S)^i,
\tag{29}
$$

where $\hat{T}_g^n(\cdot|\mu,\omega)) := \hat{T}_g(\cdot|\hat{T}_g(\cdot|...\hat{T}_g(\cdot|\mu,a_0,\omega)...,a_{n-2},\omega),a_{n-1},\omega)$, and $\hat{T}_g^n(\cdot|\mu,\hat{\omega}))$ is defined similarly.

*Proof.* We first derive the bound for one-step prediction error given Lemma E.3.

$$\xi(1) := W(\hat{T}_g(\cdot|\mu, \omega), \hat{T}_g(\cdot|\mu, \hat{\omega}))$$

$$:= \sup_f \int \int (\hat{T}(s'|s, a_0, \omega) - \hat{T}(s'|s, a_0, \hat{\omega})) f(s') \mu(s) ds ds'$$

$$\leq \int \sup_f \int (\hat{T}(s'|s, a_0, \omega) - \hat{T}(s'|s, a_0, \hat{\omega})) f(s') ds' \mu(s) ds \qquad (30)$$

$$= \int W(\hat{T}(\cdot|s, a_0, \omega), \hat{T}(\cdot|s, a_0, \hat{\omega})) \mu(s) ds$$

$$\leq \int K_{\hat{T}}^{\Omega} \epsilon \mu(s) ds = K_{\hat{T}}^{\Omega} \epsilon$$

Then, we can prove the bound for $\xi(n)$:

$$\xi(n) := W(\hat{T}_g^n(\cdot|\mu, \omega), \hat{T}_g^n(\cdot|\mu, \hat{\omega}))$$

$$\leq W(\hat{T}_g^n(\cdot|\mu, \omega), \hat{T}_g^n(\cdot|\hat{T}_g^{n-1}(\cdot|\mu, \hat{\omega}), \omega)) + W(\hat{T}_g^n(\cdot|\hat{T}_g^{n-1}(\cdot|\mu, \hat{\omega}), \omega), \hat{T}_g^n(\cdot|\mu, \hat{\omega}))$$

$$= W(\hat{T}_g^n(\cdot|\hat{T}_g^{n-1}(\cdot|\mu, \omega), \omega), \hat{T}_g^n(\cdot|\hat{T}_g^{n-1}(\cdot|\mu, \hat{\omega}), \omega)) + W(\hat{T}_g^n(\cdot|\hat{T}_g^{n-1}(\cdot|\mu, \hat{\omega}), \omega), \hat{T}_g^n(\cdot|\hat{T}_g^{n-1}(\cdot|\mu, \hat{\omega}), \hat{\omega}))$$

$$\leq K_{\hat{T}}^S W(\hat{T}_g^{n-1}(\cdot|\mu, \omega), \hat{T}_g^{n-1}(\cdot|\mu, \hat{\omega})) + K_{\hat{T}}^{\Omega} \epsilon$$

$$= K_{\hat{T}}^S \xi(n-1) + K_{\hat{T}}^{\Omega} \epsilon \leq K_{\hat{T}}^{\Omega} \epsilon \sum_{i=0}^{n-1} (K_{\hat{T}}^S)^i$$

$$(31)$$

We can get the same results if we replace $\hat{T}_g^n(\cdot|\hat{T}_g^{n-1}(\cdot|\mu, \hat{\omega}), \omega)$ with $\hat{T}_g^n(\cdot|\hat{T}_g^{n-1}(\cdot|\mu, \omega), \hat{\omega})$ in the triangle inequality. $\qquad \square$

**Lemma 4.2.** Given a HiP-MDP with learned $(K_{\hat{R}}^{\Omega}, K_{\hat{T}}^{\Omega})$-Lipschitz transition model $\hat{T}$, $\hat{R}$, in **Model Transfer**, the generalized value function is Lipschitz continuous with respect to $\Omega$ with a constant bounded by:

$$K_{V^{\pi}}^{\Omega} \leq K_{\hat{R}}^{\Omega} + \gamma K_{Q^{\pi}}^{\Omega} + \gamma K_{Q^{\pi}}^S K_{\hat{T}}^{\Omega}.$$

*Proof.* Recall that:

$$V_g^{\pi}(s) = R_g(s, a, \omega(\theta)) + \gamma \int_{s'} \hat{T}_g(s'|s, a, \omega(\theta)) Q_g^{\pi}(s, \pi(s), \omega(\theta)) ds'$$

Then we have:

$$K_{V^{\pi}}^{\Omega} = \sup_{s \in S} \sup_{\omega_1, \omega_2} \frac{|V^{\pi}(s, \omega_1(\theta)) - V^{\pi}(s, \omega_2(\theta))|}{d_{\Omega}(\omega_1, \omega_2)}$$

$$\leq \sup_{s \in S} \sup_{\omega_1, \omega_2} \frac{|\hat{R}_g(s, \pi(s), \omega_1(\theta)) - \hat{R}_g(s, \pi(s), \omega_2(\theta))|}{d_{\Omega}(\omega_1, \omega_2)}$$

$$+ \gamma \sup_{s \in S} \sup_{\omega_1, \omega_2} \frac{|\int_{s_1'} \hat{T}_g(s_1'|s, \pi(s), \omega_1(\theta)) Q_g^{\pi}(s_1', \pi(s), \omega_1(\theta)) ds_1' - \int_{s_2'} \hat{T}_g(s_2'|s, \pi(s), \omega_2(\theta)) Q_g^{\pi}(s_2', \pi(s), \omega_2(\theta)) ds_2'|}{d_{\Omega}(\omega_1, \omega_2)}$$

$$= K_{\hat{R}}^{\Omega} + \gamma \sup_{s \in S} \sup_{\omega_1, \omega_2} \frac{|\int_{s_1'} \hat{T}_g(s_1'|s, \pi(s), \omega_1) Q_g^{\pi}(s_1', \pi(s), \omega_1) ds_1' - \int_{s_2'} \hat{T}_g(s_2'|s, \pi(s), \omega_2) Q_g^{\pi}(s_2', \pi(s), \omega_2) ds_2'|}{d_{\Omega}(\omega_1, \omega_2)}$$

Recall that we quantify the distance between $\omega_1(\theta)$ and $\omega_2(\theta)$ using Wasserstein Metric $W(\omega_1, \omega_2)$, then

$$K_{V^\pi}^\Omega = K_{\hat{R}}^\Omega + \gamma \; \sup_{s \in S} \sup_{\omega_1, \omega_2} \frac{|\int_{s'_1} \hat{T}_g(s'_1|s, \pi(s), \omega_1) Q_g^\pi(s'_1, \pi(s), \omega_1) ds'_1 - \int_{s'_2} \hat{T}_g(s'_2|s, \pi(s), \omega_2) Q_g^\pi(s'_2, \pi(s), \omega_2) ds'_2|}{W(\omega_1, \omega_2)}$$

$$\leq K_{\hat{R}}^\Omega + \gamma \; \sup_{s \in S} \sup_{\omega_1, \omega_2} \frac{|\int_{s'_1} [\hat{T}_g(s'_1|s, \pi(s), \omega_1) Q_g^\pi(s'_1, \pi(s), \omega_1) - \hat{T}_g(s'_1|s, \pi(s), \omega_1) Q_g^\pi(s'_1, \pi(s), \omega_2)] ds'_1|}{W(\omega_1, \omega_2)}$$

$$+ \gamma \; \sup_{s \in S} \sup_{\omega_1, \omega_2} \frac{|\int_{s'_1} \hat{T}_g(s'_1|s, \pi(s), \omega_1) Q_g^\pi(s'_1, \pi(s), \omega_2) ds'_1 - \int_{s'_2} \hat{T}_g(s'_2|s, \pi(s), \omega_2) Q_g^\pi(s'_2, \pi(s), \omega_2) ds'_2|}{W(\omega_1, \omega_2)}$$

$$\leq K_{\hat{R}}^\Omega + \gamma \; \sup_{s \in S} \sup_{\omega_1, \omega_2} \frac{|\int_{s'_1} \hat{T}_g(s'_1|s, \pi(s), \omega_1) [Q_g^\pi(s'_1, \pi(s), \omega_1) - Q_g^\pi(s'_1, \pi(s), \omega_2)] ds'_1|}{W(\omega_1, \omega_2)}$$

$$+ \gamma \; \sup_{s \in S} \sup_{\omega_1, \omega_2} \frac{|\int_{s'_1} \hat{T}_g(s'_1|s, \pi(s), \omega_1) Q_g^\pi(s'_1, \pi(s), \omega_2) ds'_1 - \int_{s'_2} \hat{T}_g(s'_2|s, \pi(s), \omega_2) Q_g^\pi(s'_2, \pi(s), \omega_2) ds'_2|}{W(\omega_1, \omega_2)}$$

$$\leq K_{\hat{R}}^\Omega + \gamma \; \sup_{s \in S} \sup_{\omega_1, \omega_2} \frac{|\int_{s'_1} \hat{T}_g(s'_1|s, \pi(s), \omega_1) K_{Q^\pi}^\Omega W(\omega_1, \omega_2) ds'_1|}{W(\omega_1, \omega_2)}$$

$$+ \gamma \; \sup_{s \in S} \sup_{\omega_1, \omega_2} \frac{|\int_{s'_1} \hat{T}_g(s'_1|s, \pi(s), \omega_1) Q_g^\pi(s'_1, \pi(s), \omega_2) ds'_1 - \int_{s'_2} \hat{T}_g(s'_2|s, \pi(s), \omega_2) Q_g^\pi(s'_2, \pi(s), \omega_2) ds'_2|}{W(\omega_1, \omega_2)}$$

$$= K_{\hat{R}}^\Omega + \gamma K_{Q^\pi}^\Omega + \gamma \; \sup_{s \in S} \sup_{\omega_1, \omega_2} \frac{|\int_{s'_1} \hat{T}_g(s'_1|s, \pi(s), \omega_1) Q_g^\pi(s'_1, \pi(s), \omega_2) ds'_1 - \int_{s'_2} \hat{T}_g(s'_2|s, \pi(s), \omega_2) Q_g^\pi(s'_2, \pi(s), \omega_2) ds'_2|}{W(\omega_1, \omega_2)}$$

$$= K_{\hat{R}}^\Omega + \gamma K_{Q^\pi}^\Omega + \gamma \; \sup_{s \in S} \sup_{\omega_1, \omega_2} \frac{|\mathbb{E}_{s'_1 \sim \hat{T}_g(s'_1|s, \pi(s), \omega_1)} Q_g^\pi(s'_1, \pi(s), \omega_2) - \mathbb{E}_{s'_2 \sim \hat{T}_g(s'_2|s, \pi(s), \omega_2)} Q_g^\pi(s'_2, \pi(s), \omega_2)|}{W(\omega_1, \omega_2)}$$

$$\leq K_{\hat{R}}^\Omega + \gamma K_{Q^\pi}^\Omega + \gamma \; \sup_{s \in S} \sup_{\omega_1, \omega_2} \frac{K_{Q^\pi}^S W(\hat{T}_g(\cdot|s, \pi(s), \omega_1), \hat{T}_g(\cdot|s, \pi(s), \omega_2))}{W(\omega_1, \omega_2)}$$

$$\leq K_{\hat{R}}^\Omega + \gamma K_{Q^\pi}^\Omega + \gamma K_{Q^\pi}^S K_{\hat{T}}^\Omega$$

$\square$

**Theorem 4.3.** Given a HiP-MDP with learned $(K_{\hat{R}}^S, K_{\hat{T}}^S, K_{\hat{R}}^A, K_{\hat{T}}^A, K_{\hat{R}}^\Omega, K_{\hat{T}}^\Omega)$-Lipschitz transition model $\hat{T}$, $\hat{R}$, in **Model Transfer**, if $\gamma(K_T^S + K_\pi^S K_T^A) < 1$, the generalized value function corresponding to the $K_\pi^S$-Lipschitz policy $\pi(a|s)$ is Lipschitz continuous with respect to $\Omega$ with a constant bounded by:

$$K_{V^\pi}^\Omega \leq \frac{K_R^\Omega + \gamma K_{Q^\pi}^S K_T^\Omega}{1 - \gamma}. \tag{32}$$

*Proof.* By further assuming the Lipschitz continuity of dynamics & policy in Lemma 4.2, we can get the result bound leveraging the dual form of Wasserstein Metric (Equation 2), triangle inequality, and computing the fixed point of recurrence.

Recall that:

$$Q_{n+1}(s, a, \omega(\theta)) \leftarrow R_g(s, a, \omega(\theta)) + \gamma \int \hat{T}_g(s'|s, a, \omega(\theta)) V_n^\pi(s', \omega(\theta)) ds'$$

$$V_n^\pi(s', \omega(\theta)) = Q_n^\pi(s', a, \omega(\theta)), \text{ where } a \sim \pi(s)$$

It is well known that as $n \to \infty$, $Q_{n+1}$ converges to $Q^*$, now let:

$$
\begin{aligned}
K_{Q,n+1} &= \sup_{s \in S} \sup_{a \in A} \sup_{\omega_1, \omega_2} \frac{|Q_{n+1}(s, a, \omega_1(\theta)) - Q_{n+1}(s, a, \omega_2(\theta))|}{d_\omega(\omega_1, \omega_2)} \\
&\leq \sup_{s \in S} \sup_{a \in A} \sup_{\omega_1, \omega_2} \frac{|\hat{R}_g(s, a, \omega_1(\theta)) - \hat{R}_g(s, a, \omega_2(\theta))|}{d_\omega(\omega_1, \omega_2)} \\
&+ \gamma \sup_{s \in S} \sup_{a \in A} \sup_{\omega_1, \omega_2} \frac{|\int_{s_1'} \hat{T}_g(s_1'|s, a, \omega_1(\theta))Q_n^\pi(s_1', a, \omega_1(\theta))ds_1' - \int_{s_2'} \hat{T}_g(s_2'|s, a, \omega_2(\theta))Q_n^\pi(s_2', a, \omega_2(\theta))ds_2'|}{d_\omega(\omega_1, \omega_2)} \\
&= K_{\hat{R}}^\Omega + \gamma \sup_{s \in S} \sup_{a \in A} \sup_{\omega_1, \omega_2} \frac{|\int_{s_1'} \hat{T}_g(s_1'|s, a, \omega_1)Q_n^\pi(s_1', a, \omega_1)ds_1' - \int_{s_2'} \hat{T}_g(s_2'|s, a, \omega_2)Q_n^\pi(s_2', a, \omega_2)ds_2'|}{d_\omega(\omega_1, \omega_2)}
\end{aligned}
\tag{33}
$$

Recall that we quantify the distance between $\omega_1(\theta)$ and $\omega_2(\theta)$ using Wasserstein Metric $W(\omega_1, \omega_2)$, then

$$
\begin{aligned}
K_{Q,n+1} &= K_{\hat{R}}^\Omega + \gamma \sup_{s \in S} \sup_{a \in A} \sup_{\omega_1, \omega_2} \frac{|\int_{s_1'} \hat{T}_g(s_1'|s, a, \omega_1)Q_n^\pi(s_1', a, \omega_1)ds_1' - \int_{s_2'} \hat{T}_g(s_2'|s, a, \omega_2)Q_n^\pi(s_2', a, \omega_2)ds_2'|}{W(\omega_1, \omega_2)} \\
&\leq K_{\hat{R}}^\Omega + \gamma \sup_{s \in S} \sup_{a \in A} \sup_{\omega_1, \omega_2} \frac{|\int_{s_1'} [\hat{T}_g(s_1'|s, a, \omega_1)Q_n^\pi(s_1', a, \omega_1) - \hat{T}_g(s_1'|s, a, \omega_1)Q_n^\pi(s_1', a, \omega_2)]ds_1'|}{W(\omega_1, \omega_2)} \\
&+ \gamma \sup_{s \in S} \sup_{a \in A} \sup_{\omega_1, \omega_2} \frac{|\int_{s_1'} \hat{T}_g(s_1'|s, a, \omega_1)Q_n^\pi(s_1', a, \omega_2)ds_1' - \int_{s_2'} \hat{T}_g(s_2'|s, a, \omega_2)Q_n^\pi(s_2', a, \omega_2)ds_2'|}{W(\omega_1, \omega_2)} \\
&\leq K_{\hat{R}}^\Omega + \gamma \sup_{s \in S} \sup_{a \in A} \sup_{\omega_1, \omega_2} \frac{|\int_{s_1'} \hat{T}_g(s_1'|s, a, \omega_1)[Q_n^\pi(s_1', a, \omega_1) - Q_n^\pi(s_1', a, \omega_2)]ds_1'|}{W(\omega_1, \omega_2)} \\
&+ \gamma \sup_{s \in S} \sup_{a \in A} \sup_{\omega_1, \omega_2} \frac{|\int_{s_1'} \hat{T}_g(s_1'|s, a, \omega_1)Q_n^\pi(s_1', a, \omega_2)ds_1' - \int_{s_2'} \hat{T}_g(s_2'|s, a, \omega_2)Q_n^\pi(s_2', a, \omega_2)ds_2'|}{W(\omega_1, \omega_2)} \\
&\leq K_{\hat{R}}^\Omega + \gamma \sup_{s \in S} \sup_{a \in A} \sup_{\omega_1, \omega_2} \frac{|\int_{s_1'} \hat{T}_g(s_1'|s, a, \omega_1)K_{Q,n}W(\omega_1, \omega_2)ds_1'|}{W(\omega_1, \omega_2)} \\
&+ \gamma \sup_{s \in S} \sup_{a \in A} \sup_{\omega_1, \omega_2} \frac{|\int_{s_1'} \hat{T}_g(s_1'|s, a, \omega_1)Q_n^\pi(s_1', a, \omega_2)ds_1' - \int_{s_2'} \hat{T}_g(s_2'|s, a, \omega_2)Q_n^\pi(s_2', a, \omega_2)ds_2'|}{W(\omega_1, \omega_2)} \\
&= K_{\hat{R}}^\Omega + \gamma K_{Q,n} + \gamma \sup_{s \in S} \sup_{a \in A} \sup_{\omega_1, \omega_2} \frac{|\int_{s_1'} \hat{T}_g(s_1'|s, a, \omega_1)Q_n^\pi(s_1', a, \omega_2)ds_1' - \int_{s_2'} \hat{T}_g(s_2'|s, a, \omega_2)Q_n^\pi(s_2', a, \omega_2)ds_2'|}{W(\omega_1, \omega_2)} \\
&= K_{\hat{R}}^\Omega + \gamma K_{Q,n} + \gamma \sup_{s \in S} \sup_{a \in A} \sup_{\omega_1, \omega_2} \frac{|\mathbb{E}_{s_1' \sim \hat{T}_g(s_1'|s, a, \omega_1)}Q_n^\pi(s_1', a, \omega_2) - \mathbb{E}_{s_2' \sim \hat{T}_g(s_2'|s, a, \omega_2)}Q_n^\pi(s_2', a, \omega_2)|}{W(\omega_1, \omega_2)} \\
&\leq K_{\hat{R}}^\Omega + \gamma K_{Q,n} + \gamma \sup_{s \in S} \sup_{a \in A} \sup_{\omega_1, \omega_2} \frac{K_{Q^\pi}^S W(\hat{T}_g(\cdot|s, a, \omega_1), \hat{T}_g(\cdot|s, a, \omega_2))}{W(\omega_1, \omega_2)} \\
&\leq K_{\hat{R}}^\Omega + \gamma K_{Q,n} + \gamma K_{Q^\pi}^S K_{\hat{T}}^\Omega
\end{aligned}
\tag{34}
$$

Recall the bound for $(K_{\hat{R}}^S, K_{\hat{T}}^S)$-lipschitz MDP:

$$
K_{Q^\pi}^S := \sup_\omega \sup_{a \in A} \sup_{s_1, s_2} \frac{|Q(s_1, a, \omega(\theta)) - Q(s_2, a, \omega(\theta))|}{d_s(s_1, s_2)} \leq \frac{K_{\hat{R}}^S - \gamma(K_{\hat{R}}^S K_{\hat{T}}^A - K_{\hat{R}}^A K_{\hat{T}}^S)}{1 - \gamma(K_{\hat{T}}^A + K_\pi^S K_{\hat{T}}^S)}.
$$

Equivalently:

$$
K_{Q,n+1} \leq \sum_{i=0}^{n} \gamma^i K_{\hat{R}}^\Omega + \sum_{i=1}^{n+1} \gamma^i K_{Q^\pi}^S \cdot K_{\hat{T}}^\Omega + \gamma^n K_{Q,0}
\tag{35}
$$

By computing the limit of both sides:

$$K_{Q^\pi}^\Omega = \lim_{n\to\infty} K_{Q,n+1} \le \lim_{n\to\infty} \sum_{i=0}^{n} \gamma^i K_{\hat{R}}^\Omega + \lim_{n\to\infty} \sum_{i=1}^{n+1} \gamma^i K_{Q^\pi}^S \cdot K_{\hat{T}}^\Omega + \lim_{n\to\infty} \gamma^n K_{Q,0}$$

$$= \frac{K_{\hat{R}}^\Omega}{1-\gamma} + \frac{\gamma K_{Q^\pi}^S \cdot K_{\hat{T}}^\Omega}{1-\gamma} + 0$$

From Lemma 4.2 we have:

$$K_{V^\pi}^\Omega \le K_{\hat{R}}^\Omega + \gamma K_{Q^\pi}^\Omega + \gamma K_{Q^\pi}^S K_{\hat{T}}^\Omega$$

Thus,

$$K_{V^\pi}^\Omega \le \frac{K_R^\Omega + \gamma K_{Q^\pi}^S K_T^\Omega}{1-\gamma}$$

$\square$

**Lemma 4.4.** Given a $(K_R^S, K_T^S, K_R^\Omega, K_T^\Omega)$-Lipschitz HiP-MDP, for the optimal policy $\pi^*$, $|V^{\pi^*}(s, \omega_1(\theta)) - V^{\pi^*}(s, \omega_2(\theta))| \le \max_{a\in A} |Q^{\pi^*}(s, a, \omega_1(\theta)) - Q^{\pi^*}(s, a, \omega_2(\theta))|$.

*Proof.*

$$|V^{\pi^*}(s, \omega_1(\theta)) - V^{\pi^*}(s, \omega_2(\theta))| = |\max_{a\in A} Q^{\pi^*}(s, a, \omega_1(\theta)) - \max_{a\in A} Q^{\pi^*}(s, a, \omega_2(\theta))|$$

$$\le \max_{a\in A} |Q^{\pi^*}(s, a, \omega_1(\theta)) - Q^{\pi^*}(s, a, \omega_2(\theta))|$$

$\square$

**Lemma 4.5.** Given a HiP-MDP with learned $(K_{\hat{R}}^S, K_{\hat{T}}^S, K_{\hat{R}}^\Omega, K_{\hat{T}}^\Omega)$-Lipschitz transition model $\hat{T}$, $\hat{R}$, and the optimal policy $\hat{\pi}^*(a|s)$ for $\hat{T}$ and $\hat{R}$, starting from state distribution $\mu$, the expected discounted reward difference induced by hidden parameter estimation error $\epsilon$ is bounded by:

$$|J_{\pi^*}^\mu - J_{\hat{\pi}^*}^\mu| \le \frac{K_{\hat{R}}^\Omega}{1-\gamma} \cdot \epsilon + \frac{\gamma K_{\hat{R}}^S K_{\hat{T}}^\Omega}{(1-\gamma)(1-\gamma K_{\hat{T}}^S)} \cdot \epsilon, \tag{36}$$

if $\gamma K_{\hat{T}}^S < 1$.

*Proof.* Recall that:

$$J_\pi^\mu := \int_s V_g^\pi(s, \omega(\theta)) \mu(s) ds.$$

We have:

$$K_J^\Omega = \sup_s \sup_{\omega_1, \omega_2} \frac{\int_s |V^{\pi^*}(s, \omega_1(\theta)) - V^{\pi^*}(s, \omega_2(\theta))| \mu(s) ds}{d(\omega_1(\theta), \omega_2(\theta))}$$

$$\le \sup_s \sup_a \sup_{\omega_1, \omega_2} \frac{\int_s |Q^{\pi^*}(s, a, \omega_1(\theta)) - Q^{\pi^*}(s, a, \omega_2(\theta))| \mu(s) ds}{d(\omega_1(\theta), \omega_2(\theta))}$$

$$\le K_{Q^{\pi^*}}^\Omega = \lim_{n\to\infty} K_{Q,n+1}$$

$$\le \lim_{n\to\infty} \sum_{i=0}^{n} \gamma^i K_{\hat{R}}^\Omega + \lim_{n\to\infty} \sum_{i=1}^{n+1} \gamma^i K_{Q^*}^S \cdot K_{\hat{T}}^\Omega + \lim_{n\to\infty} \gamma^n K_{Q,0}$$

$$= \frac{K_{\hat{R}}^\Omega}{1-\gamma} + \frac{\gamma K_{Q^*}^S \cdot K_{\hat{T}}^\Omega}{1-\gamma} + 0$$

From Corollary 3.2, we have:

$$K_{Q^*}^S \le \frac{K_R^S}{1-\gamma K_T^S}$$

Thus,

$$K_J^\Omega \le \frac{K_{\hat{R}}^\Omega (1 - \gamma K_{\hat{T}}^S) + \gamma K_{\hat{R}}^S K_{\hat{T}}^\Omega}{(1 - \gamma)(1 - \gamma K_{\hat{T}}^S)}$$

□

**Claim 4.6.** Given a linear and deterministic transition function and reward function, the bound derived in Theorem 4.1 & Lemma 4.5 are tight.

*Proof.* Assume a linear transition function $T$ for a new task with hidden parameter $\theta$ defined as:

$$T(s, a) = As + Ba + C\theta, \tag{37}$$

And a reward function defined as:

$$R(s, a) = Ds + Ea + F\theta. \tag{38}$$

Assume there's an estimation error $\epsilon$ for $\theta$, then the transition function and reward function becomes:

$$\hat{T}(s, a) = As + Ba + C(\theta + \epsilon), \tag{39}$$

$$\hat{R}(s, a) = Ds + Ea + F(\theta + \epsilon), \tag{40}$$

First observe that:

$$\forall s, a \ |T(s, a) - \hat{T}(s, a)| = C\epsilon, \tag{41}$$

And that for $n = 2$:

$$\forall s \ |T(T(s, a_0), a_1) - \hat{T}(\hat{T}(s, a_0), a_1)| = |A(C(\epsilon + \theta) - C\theta) + Ba_1 - Ba_1 + C(\epsilon + \theta) - C\theta|$$

$$= |AC\epsilon + C\epsilon| = C\epsilon \sum_{i=0}^{1} A^i \tag{42}$$

More generally, for $n$ step compounding error of dynamics prediction $T^n$ and $\hat{T}^n$, given a fixed sequence of actions $a_0, a_1, \cdots, a_n$:

$$\forall s \ |T^n(s, a_n) - \hat{T}^n(s, a_n)| = C\epsilon \sum_{i=0}^{n} A^i \tag{43}$$

Thus, the bound in Theorem 4.1 is tight.

Now consider the state $s = 0$ and the action space only consists of one action $a = 0$ (thus the policy is optimal), we can calculate the value of $s$ predicted with $\hat{T}$ and $\hat{R}$:

$$\hat{V}(s = 0) = \hat{R}(0) + \gamma\Big(\hat{R}(0 + C(\theta + \epsilon)) + \gamma\big(\hat{R}(AC(\theta + \epsilon) + C(\theta + \epsilon)) + \cdots\big)\Big)$$

$$= F(\theta + \epsilon)\sum_{j=0}^{\infty}\gamma^j + DC(\theta + \epsilon)\sum_{n=1}^{\infty}\gamma^n\sum_{i=0}^{n-1}A^i$$

$$= \frac{F(\theta + \epsilon)}{1 - \gamma} + \frac{DC\gamma(\theta + \epsilon)}{(1 - \gamma)(1 - \gamma A)}$$

$$= \frac{(F(1 - \gamma A) + DC\gamma)(\theta + \epsilon)}{(1 - \gamma)(1 - \gamma A)} \tag{44}$$

Thus:

$$|V(s = 0) - \hat{V}(s = 0)| = \frac{F(1 - \gamma A) + DC\gamma}{(1 - \gamma)(1 - \gamma A)} \cdot \epsilon \tag{45}$$

The result matches the bound derived in Lemma 4.5. (In deterministic cases, we assume we directly estimate the value of $\theta$, so $K_T^\Omega$ & $K_R^\Omega$ becomes $K_T^\Theta$ & $K_R^\Theta$ in this case and the Wasserstein distance becomes the L1 distance $|\hat{\theta} - \theta|$.) □

**Lemma 5.1**. Given a HiP-MDP with learned $K_\pi^\Omega$-Lipschitz joint policy $\pi(a|s, \omega(\theta))$, in **Policy Transfer**, the generalized value function with respect to $\Omega$ with a constant bounded by:

$$K_{V^\pi}^\Omega \le K_{Q^\pi}^\Omega + K_{Q^\pi}^A K_\pi^\Omega.$$

*Proof.* Given a new test task, recall that:

$$V_g^{\pi(\cdot|\omega)}(s, \phi) = Q_g^{\pi(\cdot|\omega)}(s, \pi(s, \omega), \phi)$$

where we use $\phi$ to denote the true distribution $\phi(\theta)$ over hidden parameter for the current task (which should be a Dirac $\delta$-function in practice) of the environment. Then we have:

$$
\begin{aligned}
K_{V^\pi}^\Omega &= \sup_s \sup_\phi \sup_{\omega_1, \omega_2} \frac{|V_g^{\pi(\cdot|\omega_1)}(s, \phi) - V_g^{\pi(\cdot|\omega_2)}(s, \phi)|}{d(\omega_1, \omega_2)} \\
&\le \sup_s \sup_\phi \sup_{\omega_1, \omega_2} \frac{|Q_g^{\pi(\cdot|\omega_1)}(s, \pi(s', \omega_1), \phi) - Q_g^{\pi(\cdot|\omega_2)}(s, \pi(s', \omega_2), \phi)|}{d(\omega_1, \omega_2)} \\
&\le \sup_s \sup_\phi \sup_{\omega_1, \omega_2} \frac{|Q_g^{\pi(\cdot|\omega_1)}(s, \pi(s, \omega_1), \phi(\theta)) - Q_g^{\pi(\cdot|\omega_2)}(s, \pi(s, \omega_1), \phi(\theta))|}{d(\omega_1, \omega_2)} \\
&\quad + \frac{|Q_g^{\pi(\cdot|\omega_2)}(s, \pi(s, \omega_1), \phi(\theta)) - Q_g^{\pi(\cdot|\omega_2)}(s, \pi(s, \omega_2), \phi(\theta))|}{d(\omega_1, \omega_2)} \\
&\le \sup_s \sup_\phi \sup_{\omega_1, \omega_2} \frac{K_{Q^\pi}^\Omega d(\omega_1, \omega_2)}{d(\omega_1, \omega_2)} + \frac{K_{Q^\pi}^A d(\pi(\omega_1), \pi(\omega_2)) ds'}{d(\omega_1, \omega_2)} \\
&\le K_{Q^\pi}^\Omega + \frac{K_{Q^\pi}^A K_\pi^\Omega d(\omega_1, \omega_2)}{d(\omega_1, \omega_2)} \\
&\le K_{Q^\pi}^\Omega + K_{Q^\pi}^A K_\pi^\Omega
\end{aligned}
$$

$\square$

**Theorem 5.2**. Given a $(K_R^A, K_T^A, K_R^S, K_T^S)$-Lipschitz HiP-MDP, in **Policy Transfer**, the generalized value function corresponding to the pretrained $(K_\pi^S, K_\pi^\Omega)$-Lipschitz joint policy $\pi(a|s, \omega(\theta))$, if $\gamma(K_T^S + K_\pi^S K_T^A) < 1$, is Lipschitz continuous with respect to $\Omega$ with a constant bounded by

$$K_{V^\pi}^\Omega \le \frac{K_\pi^\Omega K_{Q^\pi}^A}{1 - \gamma}. \tag{46}$$

*Proof.* Similar to model transfer, if we further assume the Lipschitz continuity of dynamics & policy in Lemma 5.1, we can get the result bound leveraging the dual form of Wasserstein Metric (Equation 2), triangle inequality, and computing the fixed point of recurrence.

Recall that:

$$Q(s, \pi(s, \omega), \phi) \leftarrow R_g(s, \pi(s, \omega), \phi) + \gamma \int \hat{T}_g(s'|s, \pi(s, \omega), \phi) V^\pi(s', \phi) ds'$$

$$K_{Q^\pi}^\Omega = \sup_s \sup_\phi \sup_{\omega_1,\omega_2} \frac{|Q_{n+1}^{\pi(\cdot|\omega_1)}(s,\pi(s,\omega_1),\phi) - Q_{n+1}^{\pi(\cdot|\omega_2)}(s,\pi(s,\omega_2),\phi)|}{d(\omega_1,\omega_2)}$$

$$\leq \sup_s \sup_\phi \sup_{\omega_1,\omega_2} \frac{|Q_{n+1}^{\pi(\cdot|\omega_1)}(s,\pi(s,\omega_1),\phi) - Q_{n+1}^{\pi(\cdot|\omega_1)}(s,\pi(s,\omega_2),\phi)|}{d(\omega_1,\omega_2)} +$$

$$\sup_s \sup_\phi \sup_{\omega_1,\omega_2} \frac{|Q_{n+1}^{\pi(\cdot|\omega_1)}(s,\pi(s,\omega_2),\phi) - Q_{n+1}^{\pi(\cdot|\omega_2)}(s,\pi(s,\omega_2),\phi)|}{d(\omega_1,\omega_2)}$$

$$\leq \sup_s \sup_\phi \sup_{\omega_1,\omega_2} \frac{|Q_{n+1}^{\pi(\cdot|\omega_1)}(s,\pi(s,\omega_1),\phi) - Q_{n+1}^{\pi(\cdot|\omega_1)}(s,\pi(s,\omega_2),\phi)|}{d(\pi(\cdot|\omega_1),\pi(\cdot|\omega_2))} \cdot \frac{d(\pi(\cdot|\omega_1),\pi(\cdot|\omega_2))}{d(\omega_1,\omega_2)} +$$

$$\sup_s \sup_\phi \sup_{\omega_1,\omega_2} \frac{|Q_{n+1}^{\pi(\cdot|\omega_1)}(s,\pi(s,\omega_2),\phi) - Q_{n+1}^{\pi(\cdot|\omega_2)}(s,\pi(s,\omega_2),\phi)|}{d(\omega_1,\omega_2)}$$

$$\leq K_{Q^\pi}^A K_\pi^\Omega + \sup_s \sup_\phi \sup_{\omega_1,\omega_2} \frac{|Q_{n+1}^{\pi(\cdot|\omega_1)}(s,\pi(s,\omega_2),\phi) - Q_{n+1}^{\pi(\cdot|\omega_2)}(s,\pi(s,\omega_2),\phi)|}{d(\omega_1,\omega_2)}$$

$$(47)$$

Now let:

$$K_{Q^\pi,n+1}^* = \sup_s \sup_a \sup_\phi \sup_{\omega_1,\omega_2} \frac{|Q_{n+1}^{\pi(\cdot|\omega_1)}(s,a,\phi) - Q_{n+1}^{\pi(\cdot|\omega_2)}(s,a,\phi)|}{d(\omega_1,\omega_2)}$$

$$\leq 0 + \gamma \cdot \sup_s \sup_a \sup_\phi \sup_{\omega_1,\omega_2} \frac{\int_{s'} T_g(s'|s,a,\phi(\theta)) |Q_n^{\pi(\cdot|\omega_1)}(s',\pi(s',\omega_1),\phi) - Q_n^{\pi(\cdot|\omega_2)}(s',\pi(s',\omega_2),\phi)| ds'}{d(\omega_1,\omega_2)}$$

$$\leq \gamma \cdot \sup_s \sup_a \sup_\phi \sup_{\omega_1,\omega_2} \frac{\int_{s'} T_g(s'|s,a,\phi) |Q_n^{\pi(\cdot|\omega_1)}(s',\pi(s',\omega_1),\phi) - Q_n^{\pi(\cdot|\omega_2)}(s',\pi(s',\omega_1),\phi)| ds'}{d(\omega_1,\omega_2)}$$

$$+ \frac{\int_{s'} T_g(s'|s,a,\phi) |Q_n^{\pi(\cdot|\omega_2)}(s',\pi(s',\omega_1),\phi) - Q_n^{\pi(\cdot|\omega_2)}(s',\pi(s',\omega_2),\phi)| ds'}{d(\omega_1,\omega_2)}$$

$$\leq \gamma \cdot \sup_s \sup_a \sup_\phi \sup_{\omega_1,\omega_2} \frac{\int_{s'} T_g(s'|s,a,\phi) K_{Q^\pi,n}^* d(\omega_1,\omega_2) ds'}{d(\omega_1,\omega_2)}$$

$$+ \frac{\int_{s'} T_g(s'|s,a,\phi) K_{Q^\pi}^A d(\pi(\omega_1),\pi(\omega_2)) ds'}{d(\omega_1,\omega_2)}$$

$$\leq \gamma \cdot \sup_s \sup_a \sup_\phi \sup_{\omega_1,\omega_2} \frac{\int_{s'} T_g(s'|s,a,\phi) K_{Q^\pi,n}^* d(\omega_1,\omega_2) ds'}{d(\omega_1,\omega_2)} + \frac{\int_{s'} T_g(s'|s,a,\phi) K_{Q^\pi}^A K_\pi^\Omega d(\omega_1,\omega_2) ds'}{d(\omega_1,\omega_2)}$$

$$\leq \gamma(K_{Q^\pi,n}^* + K_{Q^\pi}^A K_\pi^\Omega)$$

Equivalently:

$$K_{Q^\pi,n+1}^* \leq \gamma^n K_{Q^\pi,0}^* + \sum_{i=1}^{n+1} \gamma^i K_{Q^\pi}^A K_\pi^\Omega$$

Computing the limit of both sides:

$$K_{Q^\pi}^* = \lim_{n\to\infty} K_{Q^\pi,n+1}^* \leq \lim_{n\to\infty} \gamma^n K_{Q^\pi,0}^* + \lim_{n\to\infty} \sum_{i=1}^{n+1} \gamma^i K_{Q^\pi}^A K_\pi^\Omega$$

$$= 0 + \frac{\gamma K_{Q^\pi}^A K_\pi^\Omega}{1-\gamma}$$

Plugging the results back in to Equation 47, we have:

$$K_{V^\pi}^\Omega = K_{Q^\pi}^\Omega \leq \frac{K_{Q^\pi}^A K_\pi^\Omega}{1-\gamma}$$

$$\square$$

**Lemma 5.3**. Given a $(K_R^A, K_T^A, K_R^S, K_T^S)$-Lipschitz HiP-MDP, a pretrained $K_\pi^\Omega$-Lipschitz optimal joint policy $\pi(a|s, \omega(\theta))$, starting from state distribution $\mu$, the expected discounted reward difference induced by hidden parameter estimation error $\epsilon$ is bounded by:

$$|J_{\pi^*}^\mu - J_\pi^\mu| \leq \frac{\gamma K_\pi^\Omega (K_R^A + \gamma K_R^S K_T^A - \gamma K_T^S K_R^A)}{(1 - \gamma)(1 - \gamma K_T^S)} \cdot \epsilon, \tag{48}$$

if $\gamma K_T^S \leq 1$.

*Proof.* When the policy is optimal, we have:

$$K_{Q^\pi, n+1}^\Omega = \sup_s \sup_a \sup_\phi \sup_{\omega_1, \omega_2} \frac{|Q_{n+1}^{\pi(\cdot|\omega_1)}(s, a, \phi) - Q_{n+1}^{\pi(\cdot|\omega_2)}(s, a, \phi)|}{d(\omega_1, \omega_2)}$$

$$\leq 0 + \gamma \cdot \sup_s \sup_a \sup_\phi \sup_{\omega_1, \omega_2} \frac{\int_{s'} T_g(s'|s, a, \phi(\theta))|Q_n^{\pi(\cdot|\omega_1)}(s', \pi(s', \omega_1), \phi) - Q_n^{\pi(\cdot|\omega_2)}(s', \pi(s', \omega_2), \phi)|ds'}{d(\omega_1, \omega_2)}$$

$$\leq \gamma \cdot \sup_s \sup_a \sup_\phi \sup_{\omega_1, \omega_2} \frac{\int_{s'} T_g(s'|s, a, \phi)|Q_n^{\pi(\cdot|\omega_1)}(s', \pi(s', \omega_1), \phi) - Q_n^{\pi(\cdot|\omega_2)}(s', \pi(s', \omega_1), \phi)|ds'}{d(\omega_1, \omega_2)}$$

$$+ \frac{\int_{s'} T_g(s'|s, a, \phi)|Q_n^{\pi(\cdot|\omega_2)}(s', \pi(s', \omega_1), \phi) - Q_n^{\pi(\cdot|\omega_2)}(s', \pi(s', \omega_2), \phi)|ds'}{d(\omega_1, \omega_2)}$$

$$\leq \gamma \cdot \sup_s \sup_a \sup_\phi \sup_{\omega_1, \omega_2} \frac{\int_{s'} T_g(s'|s, a, \phi)K_{Q^\pi, n}d(\omega_1, \omega_2)ds'}{d(\omega_1, \omega_2)}$$

$$+ \frac{\int_{s'} T_g(s'|s, a, \phi)K_{Q^\pi}^A d(\pi(\omega_1), \pi(\omega_2))ds'}{d(\omega_1, \omega_2)}$$

$$\leq \gamma \cdot \sup_s \sup_a \sup_\phi \sup_{\omega_1, \omega_2} \frac{\int_{s'} T_g(s'|s, a, \phi)K_{Q^\pi, n}d(\omega_1, \omega_2)ds'}{d(\omega_1, \omega_2)} + \frac{\int_{s'} T_g(s'|s, a, \phi)K_{Q^\pi}^A K_\pi^\Omega d(\omega_1, \omega_2)ds'}{d(\omega_1, \omega_2)}$$

$$\leq \gamma(K_{Q^\pi, n}^\Omega + K_{Q^\pi}^A K_\pi^\Omega)$$

Equivalently:

$$K_{Q^\pi, n+1}^\Omega \leq \gamma^n K_{Q^\pi, 0}^\Omega + \sum_{i=1}^{n+1} \gamma^i K_{Q^\pi}^A K_\pi^\Omega$$

Computing the limit of both sides:

$$K_{Q^\pi}^\Omega = \lim_{n\to\infty} K_{Q^\pi, n+1} \leq \lim_{n\to\infty} \gamma^n K_{Q^\pi, 0}^\Omega + \lim_{n\to\infty} \sum_{i=1}^{n+1} \gamma^i K_{Q^\pi}^A K_\pi^\Omega$$

$$= 0 + \frac{\gamma K_{Q^\pi}^A K_\pi^\theta}{1 - \gamma}$$

If the policy is optimal, we have:

$$|V^{\pi(\omega_1)}(s, \phi(\theta)) - V^{\pi(\omega_2)}(s, \phi(\theta))| \leq |\max_a Q^{\pi(\omega_1)}(s, a, \phi(\theta)) - \max_a Q^{\pi(\omega_2)}(s, a, \phi(\theta))|$$

$$\leq \max_a |Q^{\pi(\omega_1)}(s, a, \phi(\theta)) - Q^{\pi(\omega_2)}(s, a, \phi(\theta))|$$

Then for the Lipschitz constant of the expected return function, we have:

$$K_J^\Omega = \sup_s \sup_\phi \sup_{\omega_1, \omega_2} \frac{\int_s |V^{\pi(\omega_1)}(s, \phi(\theta)) - V^{\pi(\omega_2)}(s, \phi(\theta))|\mu(s)ds}{d(\omega_1(\theta), \omega_2(\theta))}$$

$$\leq \sup_s \sup_a \sup_\phi \sup_{\omega_1, \omega_2} \frac{\int_s |Q^{\pi(\omega_1)}(s, a, \phi(\theta)) - Q^{\pi(\omega_2)}(s, a, \phi(\theta))|\mu(s)ds}{d(\omega_1(\theta), \omega_2(\theta))}$$

$$= K_{Q^\pi}^\Omega$$

Using the results from Corollary 3.2, we get:

$$K_J^\Omega \leq \frac{\gamma K_\pi^\Omega (K_R^A + \gamma K_R^S K_T^A - \gamma K_T^S K_R^A)}{(1-\gamma)(1-\gamma K_T^S)}$$

$\square$

**Claim 5.4.** Given a linear and deterministic transition function, reward function and policy, the bounds derived in Lemma 5.3 are tight.

*Proof.* Assume a linear transition function $T$ for a new task defined as:

$$T(s,a) = As + Ba, \tag{49}$$

a reward function defined as:

$$R(s,a) = Ds + Ea, \tag{50}$$

And the optimal policy corresponding to hidden parameter $\theta$ defined as:

$$\pi(\cdot) = H\theta \tag{51}$$

Assume there's an estimation error $\epsilon$ for $\theta$, then the policy becomes:

$$\pi(\cdot) = H(\theta + \epsilon), \tag{52}$$

Now consider the state $s = 0$, we can calculate the value of $s$ predicted with $T$, $R$ and $\hat{\pi}$:

$$\hat{V}(s=0) = R(0) + \gamma\Big( DBH(\theta + \epsilon) + EH(\theta + \epsilon) + \gamma\big( DABH(\theta + \epsilon) + DBH(\theta + \epsilon) + EH(\theta + \epsilon) + \cdots \big)\Big)$$

$$= EH(\theta + \epsilon) \sum_{j=1}^{\infty} + DBH(\theta + \epsilon) \sum_{n=1}^{\infty} \gamma^n \sum_{i=0}^{n-1} A^i + R(0)$$

$$= \Big( \frac{\gamma EH}{1-\gamma} + \frac{DBH\gamma}{(1-\gamma)(1-\gamma A)} \Big)(\theta + \epsilon) + R(0)$$

$$\tag{53}$$

Thus:

$$|\hat{V}(s=0) - V(s=0)| = \Big( \frac{\gamma EH}{1-\gamma} + \frac{DBH\gamma}{(1-\gamma)(1-\gamma A)} \Big)\epsilon$$

$$= \frac{\gamma H(E - \gamma AE + \gamma BD)}{(1-\gamma)(1-\gamma A)} \cdot \epsilon \tag{54}$$

The result matches the bound derived in Lemma 5.3. (In deterministic cases, we assume we directly estimate the value of $\theta$, so $K_\pi^\Omega$ becomes $K_\pi^\Theta$ in this case and the Wasserstein distance becomes the L1 distance $|\hat{\theta} - \theta|$.) $\square$

# F    FURTHER CLAIM ABOUT THE ASSUMPTION $\gamma K_X^Y < 1$

Similar assumptions have been made in most of previous papers discussing Lipschitz continuity in RL (Rachelson & Lagoudakis, 2010; Pirotta et al., 2015; Asadi et al., 2018; Gelada et al., 2019). Here, we make the following claim:

**Claim F.1.** *When a HiP-MDP has a fixed horizon $N$, the assumption that $\gamma(K_T^S + K_\pi^S K_T^A) < 1$ and the assumption (when the policy is optimal) $\gamma K_T^S < 1$ are both unnecessary.*

*Proof.*

$$K_V^S = \sup_\omega \sup_{s_1,s_2} \frac{|V(s_1,\omega) - V(s_2,\omega)|}{d_S(s_1,s_2)} = \sup_\omega \sup_{s_1,s_2} \frac{|\sum_{i=0}^N \gamma^i (R_1 - R_2)|}{d_S(s_1,s_2)}$$

Firstly, if the agent is at the last step $N$ of one episode:

$$K_V^S = \sup_\omega \sup_{s_1,s_2} \frac{|\sum_{i=0}^0 \gamma^i (R_1 - R_2)|}{d_S(s_1,s_2)} \leq K_R^S$$

If the agent is at step $N-1$:

$$
\begin{aligned}
K_V^S &= \sup_\omega \sup_{s_1,s_2} \frac{|\sum_{i=0}^1 \gamma^i (R_1 - R_2)|}{d_S(s_1,s_2)} \\
&\leq \sup_\omega \sup_{s_1,s_2} \frac{|R(s_1,\pi(s_1),\omega) - R(s_2,\pi(s_2),\omega)| + \gamma|R(s_1',\pi(s_1'),\omega) - R(s_2',\pi(s_2'),\omega)|}{d_S(s_1,s_2)} \\
&\leq \sup_\omega \sup_{s_1,s_2} \frac{|R(s_1,\pi(s_1),\omega) - R(s_2,\pi(s_1),\omega)| + |R(s_2,\pi(s_1),\omega) - R(s_2,\pi(s_2),\omega)|}{d_S(s_1,s_2)}
\end{aligned}
$$

$$
\begin{aligned}
+ &\gamma \sup_\omega \sup_{s_1,s_2} \frac{|R(s_1',\pi(s_1'),\omega) - R(s_2',\pi(s_2'),\omega)|}{d_S(s_1,s_2)} \\
&\leq K_R^S + K_R^A K_\pi^S + \gamma \sup_\omega \sup_{s_1,s_2} \frac{|R(s_1',\pi(s_1'),\omega) - R(s_2',\pi(s_2'),\omega)|}{d_S(s_1,s_2)} \\
&\leq K_R^S + K_R^A K_\pi^S + \gamma \sup_\omega \sup_{s_1,s_2} \frac{|R(s_1',\pi(s_1'),\omega) - R(s_2',\pi(s_1'),\omega)| + |R(s_2',\pi(s_1'),\omega) - R(s_2',\pi(s_2'),\omega)|}{d_S(s_1,s_2)} \\
&\leq K_R^S + K_R^A K_\pi^S + \gamma(K_R^S + K_R^A K_\pi^S)\sup_\omega \sup_{s_1,s_2} \frac{|T(s_1,\pi(s_1),\omega) - T(s_2,\pi(s_2),\omega)|}{d_S(s_1,s_2)} \\
&\leq K_R^S + K_R^A K_\pi^S + \gamma(K_R^S + K_R^A K_\pi^S)(K_T^S + K_T^A K_\pi^S)
\end{aligned}
$$

If the agent is at step $N-2$, similarly, we have:

$$
K_V^S \leq K_R^S + K_R^A K_\pi^S + \gamma(K_R^S + K_R^A K_\pi^S)(K_T^S + K_T^A K_\pi^S) + \gamma^2(K_R^S + K_R^A K_\pi^S)(K_T^S + K_T^A K_\pi^S)^2
$$

By induction, if the agent is at step $N-n$, we have:

$$
K_V^S \leq (K_R^S + K_R^A K_\pi^S) \cdot \frac{1 - \gamma^n (K_T^S + K_T^A K_\pi^S)^n}{1 - \gamma(K_T^S + K_T^A K_\pi^S)}
$$

Thus, the assumption that $\gamma(K_T^S + K_\pi^S K_T^A) < 1$ is unnecessary.

Similarly, when the policy is optimal, at step $N-1$:

$$
\begin{aligned}
K_V^S &= \sup_\omega \sup_{s_1,s_2} \frac{|\sum_{i=0}^1 \gamma^i (R_1 - R_2)|}{d_S(s_1,s_2)} \\
&\leq \sup_\omega \sup_a \sup_{s_1,s_2} \frac{|R(s_1,a,\omega) - R(s_2,a,\omega)| + \gamma|R(s_1',a,\omega) - R(s_2',a,\omega)|}{d_S(s_1,s_2)} \\
&\leq K_R^S + \gamma \sup_\omega \sup_a \sup_{s_1,s_2} \frac{|R(s_1',a,\omega) - R(s_2',a,\omega)|}{d_S(s_1,s_2)} \\
&\leq K_R^S + \gamma K_R^S \sup_\omega \sup_a \sup_{s_1,s_2} \frac{|T(s_1,a,\omega) - T(s_2,a,\omega)|}{d_S(s_1,s_2)} \\
&\leq K_R^S + \gamma K_R^S K_T^S
\end{aligned}
$$

At step $N-n$, we have:

$$
K_V^S \leq K_R^S \cdot \frac{1 - \gamma^n (K_T^S)^n}{1 - \gamma K_T^S}
$$

Thus, the assumption that $\gamma K_T^S < 1$ is unnecessary.

Intuitively, for tasks with infinite horizons, the assumption requires that the future states generated by close states are not too divergent. The threshold of "divergent" depends on how farsighted the agent is. $\qquad\square$

## G  APPLICATIONS OF HIP-MDPS

HiP-MDP is an important setting widely used in recent meta RL papers. It provides a natural testbed for meta/lifelong RL algorithms as the difference between tasks can be controlled by a low-dimensional latent vector. A commonly-used meta-RL benchmark (almost in every recent meta-RL

papers) creates a set of tasks by changing the environmental parameters (e.g. mass, damping) (Lee et al., 2020; Raileanu et al., 2020; Fu et al., 2022) or reward functions (e.g. target position, target velocity) (Rakelly et al., 2019; Zintgraf et al., 2020) of mujoco-simulated robots. Moreover, in real-world robotic applications, we want the robot to learn a policy that is robust to small dynamics changes like unexpected perturbations. Then one of the solutions is to let the agent pretrain over a set of HiP-MDPs considering the possible dynamics changes (Nagabandi et al., 2019).

