# OpenReview forum: "Performance Bounds for Model and Policy Transfer in Hidden-parameter MDPs"
_ICLR.cc/2023/Conference — ICLR 2023 poster_

### Official Review · Reviewer_zeh4 · 2022-10-16

**Confidence:** 3
**Correctness:** 4
**Technical Novelty And Significance:** 2
**Empirical Novelty And Significance:** Not applicable
**Recommendation:** 5

**Clarity, Quality, Novelty And Reproducibility:**

The paper writing is clear, the proof looks correct and the experiments seems easy to produce. My main concern is that the novelty of this paper is not enough.

**Strength And Weaknesses:**

### Strength

The Lipschitz continuty assumptions about the model/policy are reasonable, from theoretical perspective. It's also good to see the theoretical results are verified by empirical evaluation in Sec. 6.

### Weakness

Since the experiment part is just some evaluation about the theoretical results, instead of proposing some novel and effective empirical algorithms inspired from theory, I would believe the novelty of theory part is what to be evaluated the most.

However, it seems to me the theory part is not significant enough, and the results, although somewhat new, are kind of trivial and not surprising. Theoretical results about value difference bounds w.r.t. Lipschitz constants had been established in previous works (e.g. [1]), and the authors main contributions seems just analyzed the extended setting called HiP-MDPs, where the model/reward functions will take an additional ``hidden parameter'' $\theta$ as input which capturing the variation across different models. I found the techniques are similar and there is not much novelty.

Besides, the theory also has some restriction in practice. Although similar assumptions occurs in previous papers, the requirement that $\gamma$ times (combination of) Lipschitz constants is less than 1 occurs in almost all the main theorems (seems because of the $1-\gamma K$ occurs in the denominator). However, in practice, it is hard to verify whether such condition holds, or give any prediction about what will happen if such conditions do not hold. I think the authors should provide at least some examples about when such assumptions can be true in practice.


[1] DeepMDP: Learning Continuous Latent Space Models for Representation Learning. Carles Gelada, Saurabh Kumar, Jacob Buckman, Ofir Nachum, Marc G. Bellemare

**Summary Of The Paper:**

This paper studied the performance bound when transferring model or policy between HiP-MDPs. The authors introduced some assumptions on the model/policy Lipschitz continuty and established the Lipschitz continuty property of (optimal) value functions in HiP-MDPs. In Sec. 4 & 5, they derived the upper bound when transferring the optimal policy in learned Lipschitz model and pretrained Lipschitz optimal joint policy, repsectively. Finally, in Sec. 6, they conducted some evaluation experiments to verify the theoretical finding.

**Summary Of The Review:**

This paper studied an interesting setting. However, as I discussed in ``Weakness'' section, I think this paper is lack of novelty and the contribution is not enough for ICLR.

---

> ### Author Response · Authors · 2022-11-11
> **Initial Response to Reviewer zeh4 - Part 1**
>
> Thank you for the thoughtful review; we believe we can resolve your concerns.
>
> > “However, it seems to me the theory part is not significant enough, and the results, although somewhat new, are kind of trivial and not surprising. Theoretical results about value difference bounds w.r.t. Lipschitz constants had been established in previous works (e.g. [1]), and the authors main contributions seems just analyzed the extended setting called HiP-MDPs, where the model/reward functions will take an additional ``hidden parameter'' θ as input which capturing the variation across different models. I found the techniques are similar and there is not much novelty.”
>
> A:
>
> a. To analyze RL algorithms that transfer knowledge from previous tasks, one of the core questions is how different tasks are related, and here we are making the HiP-MDP assumption. We want to first emphasize that **HiP-MDP is an important setting widely used in recent meta RL papers**. It provides a natural testbed for meta/lifelong RL algorithms as the difference between tasks can be controlled by a low-dimensional latent vector. A commonly-used meta-RL benchmark (almost in every recent meta-RL papers) creates a set of tasks by changing the environmental parameters (e.g. mass, damping)[2, 3] or reward functions (e.g. target position, target velocity)[4, 5] of mujoco-simulated robots. Moreover, in real-world robotic applications, we want the robot to learn a policy that is robust to small dynamics changes like unexpected perturbations. Then one of the solutions is to let the agent pretrain over a set of HiP-MDPs considering the possible dynamics changes [6].
>
>
> b. The value difference bounds for HiP-MDP (Theorem 3.1 and Corollary 3.2) mentioned by the reviewer **are only one small part of our contributions**. Our main contribution includes a rigorous theoretical analysis of the policy & model transfer algorithms separately in Lipschitz HiP-MDPs, including Theorem 4.1, 4.3, 5.2 and Lemma 4.5, 5.3 in Section 4 & 5.  We derive the regret upper bounds of model and policy transfer algorithms, respectively. We also give an upper bound for multi-step prediction error in model transfer. We show that these bounds are tight in linear deterministic settings. As far as we are aware, **these are the first theoretical results about policy and model transfer performance bounds, or more generally, RL algorithms’ performance bounds in HiP-MDPs.** Even in other transfer learning settings (other than HiP-MDP), few works analyze **both** policy and model transfer in a **unified** theoretical framework, yet we also provide direct empirical performance comparison of them in a well-controlled setting besides that.
>
> Our results characterize when model/policy transfer can be more robust than the other: a slower increase in the Lipschitz constant with respect to the hidden parameter implies more robustness. Given these bounds, one direct implication to HiP-MDP algorithms in practice is that we can infer the performance trend of policy transfer and model transfer algorithms by either qualitatively or quantitatively estimating the dynamics and reward functions' sensitivity to states and actions. This can further help us determine which method is probably more advantageous for a specific HiP-MDP problem. In particular, **policy transfer is likely to outperform model transfer in the sense of robustness when the differences between actions are small, while model transfer is likely to outperform policy transfer in terms of robustness when the differences between neighboring states are small, both of which are validated by our empirical results.** As also mentioned by Reviewer qyA5, these provided practical perspectives on choosing between model or policy transfer methods. Thus, we believe these are important and novel results.

---

> > ### Author Response · Authors · 2022-11-11
> > **Initial Response to Reviewer zeh4 - Part 2**
> >
> > c. The value difference bounds are the fundamental quantities to be analyzed if one makes the Lipschitz continuity assumption in MDPs. It’s very difficult to bypass this and theoretically analyze an algorithm’s performance. Therefore our work definitely will share some similarity with previous work in this sense. Furthermore, our derived value difference bounds for HiP-MDP (Theorem 3.1 and Corollary 3.2) can directly apply to a Lipschitz regular MDP with a Lipschitz policy, as the Lipschitz continuity assumptions with respect to hidden parameters are not used in both the final results and proofs.  Yet in this case, our derived bounds **are more general compared to previous results (including the result in DeepMDP[1]) since we require fewer assumptions** about either the model or the policy. We have provided insights about relationships of this theorem with previous results for Lipschitz regular MDP works in Section 3 (the paragraph under Corollary 3.2).Our results are more general compared to prior works as (1) we do not assume that the Lipschitz constants for state and action are the same because this is usually not the case in practice; (2) we do not assume the policy is optimal: our theory applies to all Lipschitz policies. **In appendix D of the updated version of our submission**, we have also included a table showing the comparison of Theorem 3.1 with the previous theories.
> >
> > d. The DeepMDP paper mentioned by the reviewer is significantly different from our paper. 1. It is true that both papers derive a Lipschitz value difference bound. But as we explained in the last paragraph, in regular MDP settings our derived bound is more general than the one in DeepMDP as we do not make the assumption that the policy is optimal. 2. Based on the value difference bound, DeepMDP focuses on learning a latent representation model, while our paper focuses on the theoretical analysis of policy and model transfer algorithms, which is a completely different goal and setting. Moreover, as explained in b., the main results in other parts are completely different. Therefore, we respectfully disagree with the reviewer that the techniques are similar.
> >
> > > “Besides, the theory also has some restriction in practice. Although similar assumptions occurs in previous papers, the requirement that γ  times (combination of) Lipschitz constants is less than 1 occurs in almost all the main theorems (seems because of the 1−γK occurs in the denominator). However, in practice, it is hard to verify whether such condition holds, or give any prediction about what will happen if such conditions do not hold. I think the authors should provide at least some examples about when such assumptions can be true in practice.”
> >
> > A: We appreciate the reviewer for pointing out this. As a practical example, **we add explanations and proofs in appendix F of the updated version of our paper that our theorems hold whenever the horizon of the task is finite, without need to explicitly check whether this assumption holds or not.** We believe this is a reasonable and common case that can be applied to in many practical deep RL benchmarks. Besides, we note that this assumption in Theorem 3.1 is the exact same one as in [7] and [8]. Further, when the policy is optimal, the assumption becomes $\gamma K_{T}^{S} < 1$ (in Corollary 3.2, Lemma 4.5 and 5.3). This is also the exact same assumption that has been made in the DeepMDP paper [1] (which also assumes the policy is optimal) as well as in [9].  In all the cases, we did not make a stricter assumption compared to previous works. Intuitively, for tasks with infinite horizons, this assumption requires that the future states generated by close states are not too divergent. The threshold of “divergent” depends on how farsighted the agent is. This is not a very strict requirement in practice (e.g. for mujoco-based robotics tasks). In HiP-MDP settings, we do not ask more, as it can be easily observed that the assumption is not related to the Lipschitz continuity with respect to the hidden parameter.

---

> > > ### Author Response · Authors · 2022-11-11
> > > **Initial Response to Reviewer zeh4 - references**
> > >
> > > [1] DeepMDP: Learning Continuous Latent Space Models for Representation Learning. Carles Gelada, Saurabh Kumar, Jacob Buckman, Ofir Nachum, Marc G. Bellemare.http://proceedings.mlr.press/v97/gelada19a/gelada19a.pdf
> > >
> > > [2] Context-aware dynamics model for generalization in model-based reinforcement learning. Kimin Lee, Younggyo Seo, Seunghyun Lee, Honglak Lee, and Jinwoo Shin.http://proceedings.mlr.press/v119/lee20g.html
> > >
> > > [3] Fast Adaptation to New Environments via Policy-Dynamics Value Functions. Roberta Raileanu, Max Goldstein, Arthur Szlam, Rob Fergus. http://proceedings.mlr.press/v119/raileanu20a.html
> > >
> > > [4] Efficient off-policy meta-reinforcement learning via probabilistic context variables. Kate Rakelly, Aurick Zhou, Chelsea Finn, Sergey Levine, and Deirdre Quillen. http://proceedings.mlr.press/v97/rakelly19a.html
> > >
> > > [5] Varibad: A very good method for bayes-adaptive deep RL via meta-learning. Luisa M. Zintgraf, Kyriacos Shiarlis, Maximilian Igl, Sebastian Schulze, Yarin Gal, Katja Hofmann, and Shimon Whiteson. https://openreview.net/forum?id=Hkl9JlBYvr
> > >
> > > [6] Learning to adapt in dynamic, real-world environments through meta-reinforcement learning. Anusha Nagabandi, Ignasi Clavera, Simin Liu, Ronald S. Fearing, Pieter Abbeel, Sergey Levine, and Chelsea Finn. https://openreview.net/forum?id=HyztsoC5Y7
> > >
> > > [7] On the locality of action domination in sequential decision making. Emmanuel Rachelson and Michail G. Lagoudakis. http://gauss.ececs.uc.edu/Workshops/isaim2010/papers/lag.pdf
> > >
> > > [8] Policy gradient in lipschitz markov decision processes. Matteo Pirotta, Marcello Restelli, and Luca Bascetta. https://doi.org/10.1007/s10994-015-5484-1
> > >
> > > [9] Lipschitz continuity in model-based reinforcement learning. Kavosh Asadi, Dipendra Kumar Misra, and Michael L. Littman. https://arxiv.org/abs/1804.07193

---

> > > > ### Comment · Reviewer_zeh4 · 2022-11-15
> > > > **Post Rebuttal**
> > > >
> > > > Thanks for the response by authors, which detailedly highlights the interesting part of this paper and address some of my concerns. I decide to increase my score a little bit.
> > > >
> > > > I admit that this paper has some results haven't be published before, e.g., ''the first theoretical results about policy and model transfer performance bounds in Hid-MDP'', and also provides some interesting observations. However, as a theory paper, the techniques in this paper to obtain final results is quite elementary to me and share much similarity in previous papers, and therefore, lack of technique novelty. So I will keep my score to be marginal level.

---

> > > > > ### Author Response · Authors · 2022-11-15
> > > > > **Thank you for your response**
> > > > >
> > > > > We really appreciate the reviewer for taking our rebuttal into account and raising their rating of the paper. Thank you!

---

### Official Review · Reviewer_g5Zd · 2022-10-25

**Confidence:** 3
**Correctness:** 4
**Technical Novelty And Significance:** 3
**Empirical Novelty And Significance:** 3
**Recommendation:** 8

**Clarity, Quality, Novelty And Reproducibility:**

This paper has very clear writing and very high quality. The results of this paper are very solid, and I believe the experiments in this paper are reproducible. To the best of my knowledge, the novelty of the results is high.

**Details Of Ethics Concerns:**

There is no ethics concern for this paper.

**Strength And Weaknesses:**

Strength:
(1) Very clear paper writing and very high paper quality. The literature review is very complete.
(2) Proofs in this paper are very clear and easy to follow.
(3) The experiments are very complete. I believe they are reproducible.

Weakness:
The context width is quite tricky in the supplement.

**Summary Of The Paper:**

In this paper, the authors studied the Hidden-Parameter MDP (HiP-MDP) framework, which is kind of non-stationary MDP. There is a set of parameters controlling the dynamics and reward, which may vary by tasks. This setting is very important since it can be applied in meta reinforcement learning and lifelong reinforcement learning. The authors characterized the robustness of both model transfer and policy transfer algorithms with respect to the hidden parameter estimation error, which I believe is a meaningful research. In this paper, the authors not only obtained the Lipschitz continuity, but also the regret bound under both settings.

**Summary Of The Review:**

To sum up, I would like to give this paper an "accept" mainly because of its theoretical significance and novelty.

---

> ### Author Response · Authors · 2022-11-11
> **Initial Response to Reviewer g5Zd**
>
> Thank you for the very positive comments regarding the theoretical novelty and the overall quality of the paper! We are happy to address any further questions you may have. We did not understand your comment on the context width, could you please clarify?

---

### Official Review · Reviewer_qyA5 · 2022-10-28

**Confidence:** 2
**Correctness:** 3
**Technical Novelty And Significance:** 3
**Empirical Novelty And Significance:** 2
**Recommendation:** 6

**Clarity, Quality, Novelty And Reproducibility:**

The main concepts, methodology and the interpretations of the Theorems are clearly explained. The paper provides a novel theoretical analysis that helps to choose between different models in practice.

**Strength And Weaknesses:**


The strength of this paper is mainly theoretical, it rigorously characterizes the robustness of model/policy transfer methods by hidden parameter estimation error. Thus, it provides practical perspectives on choosing between model or policy transfer methods. For example, policy transfer is likely to outperform model transfer in the sense of robustness when the differences between actions are small. On the other hand, model transfer is likely to outperform policy transfer in terms of robustness when the different between neighboring states are small. Empirical results further validates the theoretical findings.

The main weaknesses of this work are listed as follows:
- It is unclear how the analysis presented in this work is related with previous theories. In many places the authors have mentioned that their results share similarity with some of the previous results, for example Theorem 3.1 is a generalization of theorems in a bunch of papers, similar paradigm of the algorithm framework has been widely used in complex continuous control tasks. The authors emphasis the major significance of their work, but it seems that it fails to provide a more detailed comparison on how much it improves upon previous results on HiP-MDPs.
- There is little intuition on how the results are derived, a brief outline of proof on at least one of the main theorems would help readers to better understand the technical novelty of this work.

**Summary Of The Paper:**

This paper considers the HiP-MDPs where a set of low-dimensional hidden parameters models the variations in tasks, this setting is naturally applicable to settings with transfers. The main contribution of this paper is the theoretical analysis on the regret upper bounds of model and policy transfer algorithms, as well as multi-step prediction error. These bounds are proven to be tight in linear deterministic settings. The theoretical analysis in this paper provides insights on the performance of model/policy transfer algorithms and its connection with Lipschitz constants.


**Summary Of The Review:**

Overall, this paper is strong in theory and the experimental results support the main theoretical claim. The presentation can be improved at some point.

---

> ### Author Response · Authors · 2022-11-11
> **Initial Response to Reviewer qyA5**
>
> Thank you for the very positive comments and thoughtful suggestions! We address the reviewer’s concerns below.
>
> > “It is unclear how the analysis presented in this work is related with previous theories. In many places the authors have mentioned that their results share similarity with some of the previous results, for example Theorem 3.1 is a generalization of theorems in a bunch of papers”:
>
> A:  First, we would like to make it clear that for the main theorems derived in this paper, only Theorem 3.1 shares certain similarity with previous results, while Theorem 4.1, 4.3, 5.2 and Lemma 4.5, 5.3 are independent of previous work —  **as far as we are aware, these are the first theoretical results about policy and model transfer performance bound in HiP-MDPs.** As for Theorem 3.1, our results generalize prior works in regular MDPs. We have provided insights about relationships of this theorem with previous results for Lipschitz regular MDP works in Section 3 (the paragraph under Corollary 3.2 highlighted in red). Specifically,  note that theorem 3.1 and Corollary 3.2 directly apply to a Lipschitz regular MDP with a Lipschitz policy, as the Lipschitz continuity assumptions with respect to hidden parameters are not used in both the final results and proofs. For regular MDPs, our results are more general compared to prior works as (1) we do not assume that the Lipschitz constants for state and action are the same because this is usually not the case in practice; (2) we do not assume the policy is optimal: our theory applies to all Lipschitz policies. We have updated appendix D to include a table showing the comparison of Theorem 3.1 with previous theories.
>
> > “similar paradigm of the algorithm framework has been widely used in complex continuous control tasks”:
>
> A: We believe the reviewer is referring to the claim we made at the beginning of Section 4. If so, we would like to emphasize that the goal of this paper is not to propose a new algorithm, instead we analyze two main categories of transfer RL algorithms (policy and model transfer) in HiP-MDPs based on the Lipschitz continuity assumption. This sentence was meant to make it clear that the algorithm framework we analyzed is commonly used in recent deep RL works, which makes our performance analysis important.
>
> > “The authors emphasis the major significance of their work, but it seems that it fails to provide a more detailed comparison on how much it improves upon previous results on HiP-MDPs.”:
>
> A: As far as we are aware, our paper is the first one to derive theoretical performance bounds for RL algorithms in HiP-MDPs, which should be considered an important aspect of our novelty.  Thus, there are not previous theoretical results on HiP-MDPs to compare to.
>
> > “There is little intuition on how the results are derived, a brief outline of proof on at least one of the main theorems would help readers to better understand the technical novelty of this work.”
>
> A: We appreciate the reviewer for pointing out this. We have added some proof sketches in the main text (highlighted in blue, in particular, Theorem 3.1, as it is the basic theorem for the other results followed) as well as in the appendix. Due to space limitations, we were not able to include the proof sketches of more theorems in the main text.

---

### Decision · Program_Chairs · 2023-01-20

**Decision:**

Accept: poster

**Justification For Why Not Higher Score:**

The theoreticals result while reassuring do not provide any new theoretical or empirical insight, and the experiments are also preliminary.

**Justification For Why Not Lower Score:**

NA

**Metareview: Summary, Strengths And Weaknesses:**

This paper provides theoretical justification for two families of methods, namely model-transfer and policy-transfer, that perform transfer learning in hidden-parameter MDPs. All reviewers agree that the theoretical results are significant and sound. It's also encouraging to see theory papers having preliminary experiments to justify the algorithm and settings. I vote for acceptance of this paper.

**Note From Pc:**

if the above contains the word "oral" or "spotlight" please see: "oral" presentation means -> notable-top-5% and "spotlight" means -> notable-top-25%. As stated in our emails, we are disassociating presentation type from AC recommendations